# Anatomy of a post-subduction collision

**Ebru Şengül Uluocak** [1,2] ✉, **Russell N. Pysklywec** [3], **Claudio Faccenna** [2,4] & **Taylor Schildgen** [2,5]

The spatiotemporal interplay of long-lasting post-orogenic deformations make continental collision zones among Earth's most enigmatic systems. Here, we employ three-dimensional thermomechanical modeling to decode mantle dynamics of the Arabian-Eurasian collision—an archetype of post-subduction tectonics. Our key findings reveal that plumelet-plate interactions drive deformation both within and at the margins of convergent plates, forming modern kinematics, surface tectonics, and plate boundary configurations. We document previously unrecognized segmentation of the subducted Neo-tethyan slabs (Bitlis and Zagros) accompanied by upper-plate tearing, which fundamentally modifies the seismotectonic stress accumulation along the Arabian-Turkish-Iranian boundary. The convective support from the plumelet beneath the former Tethyan magmatic arc drives drip-like lithospheric removal under the southern Georgian highland, providing a regional-scale example of arc-to-intraplate deformation transformation. Our results offer a unified framework for understanding how upper mantle processes control surface deformation in post-subduction systems dominated by plumelet dynamics.

Plate tectonic collision zones can experience prolonged crustal and upper mantle deformation that defines post-subduction tectonics in continental orogenic settings. Geodynamic interpretations of these processes, such as slab break-off (e.g., Eastern Anatolia[1]), mantle indentation (e.g., Alps[2]), relamination (e.g., Zagros[3]) and dripping of continental and/or oceanic lithospheric mantle and mafic lower crust (e.g., Central Anatolia[4]; Colorado Plateau[5]; Andean system[6–9]), polarity reversal of plate consumption[10] accompanied by crustal shortening and surface uplift have been studied globally (ref. [11] and references therein). However, an important unresolved aspect of post-subduction tectonics lies in deciphering the multiscale anatomy of upper mantle dynamics and its surface manifestations. This persistent knowledge gap stems from the spatiotemporal overlap of collisional processes, which produces complex, often ambiguous magmatic, tectonic, and geophysical fingerprints.

In this work, we focus on the Arabian-Eurasian collision region (Fig. 1) as a plate collision archetype where a multitude of post-subduction mantle dynamics are actively shaping the surface geology and tectonic configuration. Based on seismic imaging[12] and our geodynamic model of the upper mantle structures, we resolve the three-dimensional (3D) mantle circulation patterns with large- and small-scale convective cells, and their dynamic interactions with overlying plates. Different from previous studies, including numerical experiments in this part of the collision belt (Fig. 1b, e.g., refs. [13,14], ref. [15] and references therein), our 3D thermomechanical model, extending east to the Caspian Sea, is sensitive to both large (e.g., plumelet[15]) and regional-scale changes, which may correspond to specific zones with thinned/missing lithospheric mantle (e.g., the proposed delamination-modified regions of eastern Anatolia and western Greater Caucasus), high plateaus (e.g., Turkish-Georgian-Armenian-TGA, the East Anatolian and the NW Iranian plateaus), deep basins (e.g., Mesopotamian-Zagros, Kura, eastern Black and western Caspian seas basins) and plate boundaries (e.g., Bitlis-Zagros suture and fold-thrust belt) (Fig. 1).

The Arabian-Eurasian collision zone involves a complex interplay of crustal tectonic processes and lithospheric variations. Widespread Neogene-Quaternary volcano-sedimentary rocks occur along the col-lisional front, young volcanic centers, and high plateaus (-2 km, TGA

[1]Department of Geophysical Engineering, Çanakkale Onsekiz Mart University, Çanakkale, Türkiye. [2]GFZ German Research Centre for Geosciences, Potsdam, Germany. [3]Department of Earth Sciences, University of Toronto, Toronto, ON, Canada. [4]Department of Science, Roma Tre University, Rome, Italy. [5]Institute for Geosciences, University of Potsdam, Potsdam, Germany. ✉e-mail: ebrusengul@gmail.com

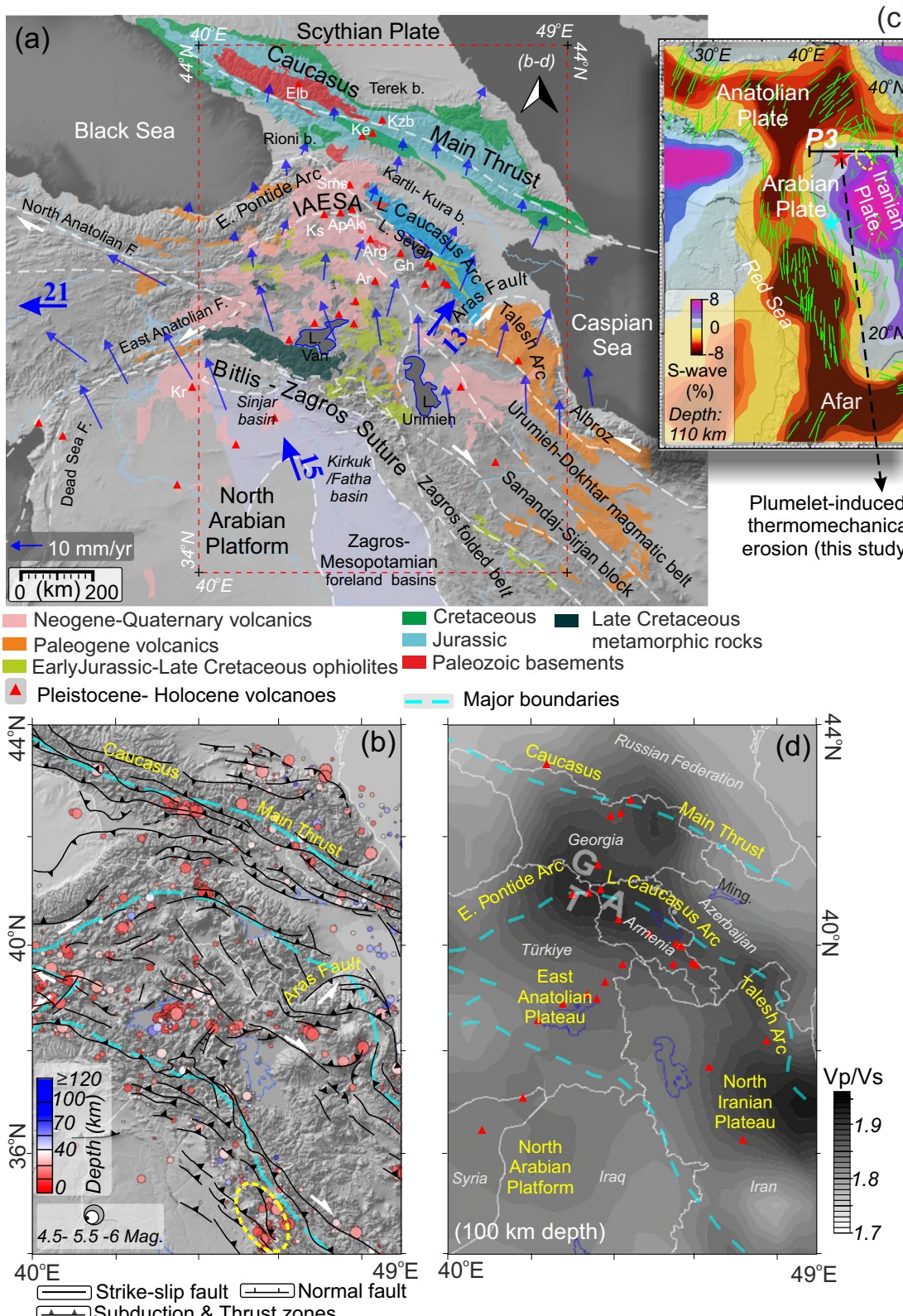

**Fig. 1 | Main structures and units of the collision zone. a** Major geological and structural variations with GPS vectors (blue arrows with average values in mm/yr[71]). IAESA İzmir-Ankara-Erzincan-Sevan-Akera suture zones. Triangles in (**a**) and (**d**) are Holocene and Pleistocene volcanoes (Ks Kısır, Ap Arpacay, Ak Akhuryan, Sms Samsari, Ar Ağrı, Arg Aragats, Gh Gegham, Elb Elbrus, Kzb Kazbek, Ke Keli[72]). The red rectangular shows the modeled area. **b** Faults, a fold-thrust system with earthquake distributions (global database between 1960 and 2024; http://www. geomapapp.org[73]). T; Kars-Ağrı-Erzurum provinces (northeastern Türkiye), G; Dzhavakheti- Javakheti provinces (southern Georgia), A; Shamiram-Ghegham provinces (southwest Armenia), E; Eastern, L; Lesser, F; Fault (major units and boundaries are derived from refs. 15,20,74–76. **c** Large-scale variations of seismic data with anisotropy pattern (ref. 31 and references therein). **d** Vp/Vs ratios for the depth of 100 km (modified from refs. 49,50). Rivers and country borders are shown with blue (**a**) and white lines (**b**, **d**), respectively.

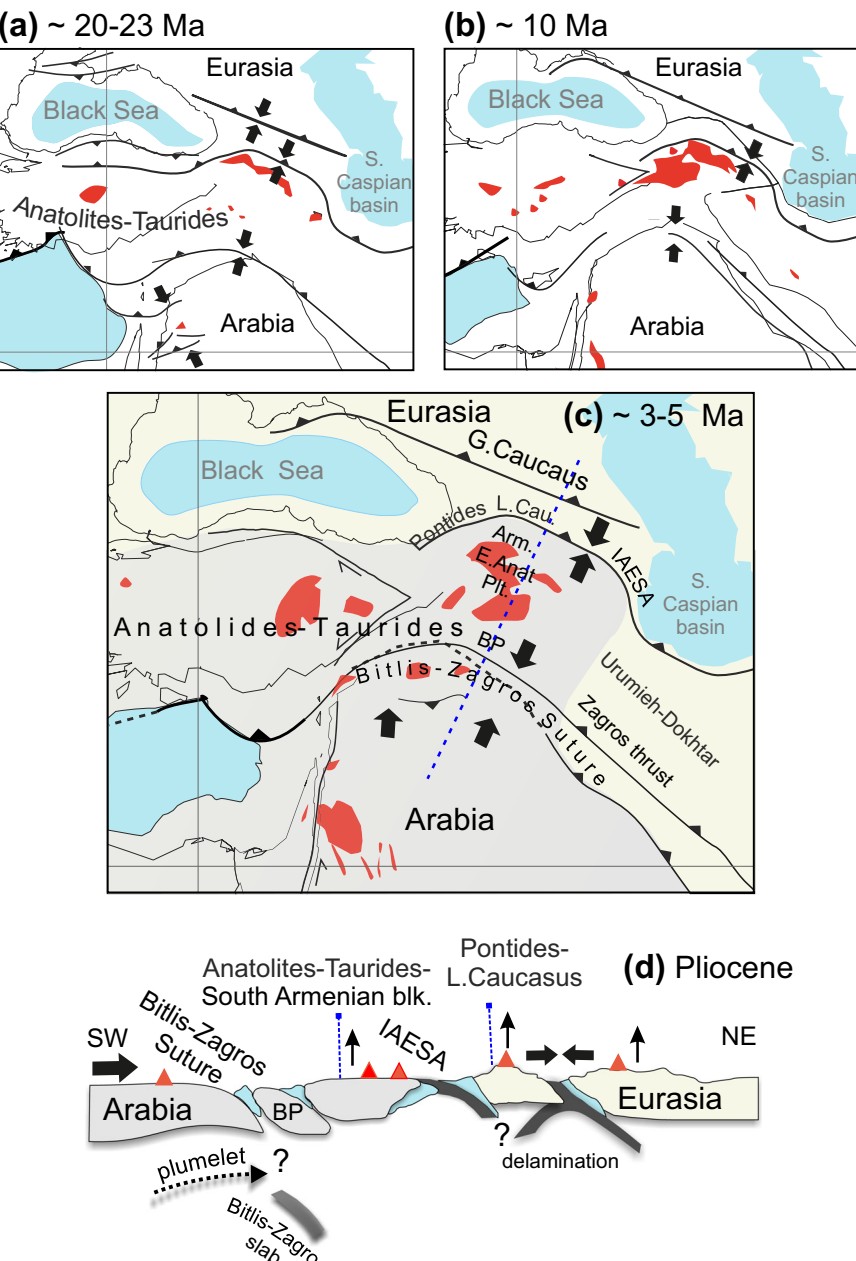

**Fig. 2 | Schematic model illustrating the previously proposed regional tectonic evolution for the Arabia-Anatolia-Caucasus domain from the Middle Miocene to the Pliocene (derived from ref. 15, ref. 20 and references therein, ref. 77 and references therein). a** Middle Miocene, **b** Late Miocene, and **c, d** Pliocene configurations. Tectonic evolution with simplified boundaries, blocks (gray and yellow areas) and proposed deformations along the profile shown in (**b**) (blue dashed line). IAESA İzmir-Ankara-Erzincan-Sevan-Akera suture zones, E Anat. Plt. East Anatolian Plateau, Arm. South Armenian block, Cau. Caucasus, L. Lesser, G. Greater, BP Bitlis-Pötürge massif (black arrows denote lateral and vertical plate movements; volcanics are marked by red zones and triangles; basins are highlighted in blue).

and North Iranian Plateau) surrounded by suture and fold-thrust belts define major physiographic units of the region (Fig. 1). Syn and post-collisional magmatic rocks with arc-type geochemical compositions extend from the Eastern Pontide and the Lesser Caucasus arcs to the south, along the Alborz magmatic belt (refs. 16,17, Fig. 1a). The northeastern Arabian margin is dominated by multi-detachment fold-thrust systems (i.e., the Zagros-Sanandaj-Sirjan basal fold and thrust belt), deforming a thick (~10–12 km) sedimentary cover and underlying crust (ref. 18, Fig. 1b). North of the intramountain Cenozoic foreland basins of the Transcaucasia (e.g., Terek, Rioni, Kartli and Kura basins), the Greater Caucasus orogen accommodates the NW-SE striking Caucasus Main Thrust (e.g., ref. 19 and references therein). Within this

regional context, a transform fault, the Aras (Araks) Fault (Fig. 1b), separates a region undergoing mostly strike-slip deformation to the west from an area experiencing shortening to the east (Fig. 2), implying a segmentation of collisional tectonics between the eastern Anatolian, eastern Greater Caucasus, and western Iranian regions (ref. 20 and references therein). However, the active mantle root of this complex collisional system remains poorly quantified.

Sub-crustal seismicity (≥~40 km depth) is predominantly recorded in the central and eastern Greater Caucasus, Kura Basin, and south-central Caspian Sea basins[20–22], (Fig. 1b). These events are commonly attributed to the NE-dipping lithospheric root of the Kura Basin and/or the remnant subducting oceanic slab beneath the eastern Greater

Caucasus[21–25] (ref. 26 and references therein). Fast seismic velocity anomalies in the large and regional-scale seismic studies image the thick lithospheric root (~200 km depth) of the Kura Basin and the eastern part of the Lesser Caucasus (e.g., refs. 12,23,24). Similarly, fast seismic velocities beneath the Terek (~4.7 km sedimentary thickness) and Caspian basins reflect dense crustal and lithospheric roots underlying thick sedimentary sequences[25,27]. Comparable high-velocity anomalies at shallow depths (≤-150 km) are observed beneath the Black Sea, Mesopotamian-Zagros foreland basins, and Zagros fold-thrust belt, consistent with high Pn seismic wave velocity perturbations in these regions[14,15,23,28,29].

In contrast to high amplitudes of guided seismic waves (Pn and Sn) that travel in the lithospheric mantle, crustal seismicity and slow seismic velocities correlate well with zones of high Sn and Pn attenuation, indicating thinned or absent lithosphere (≤50–90 km) beneath the East Anatolian, TGA, the North Iranian plateaus and the West Greater Caucasus[14,21,24,28,30]. Concerning lithospheric mantle structures with numerous lines of geophysical and petrographic evidence, including a large-scale low-velocity zone extending from Afar to the Greater Caucasus (≤-300 km depth, Fig. 1c, e.g., ref. 31), the latest numerical model by Uluocak et al.[15] implies a lithospheric channel with SW-NE mantle flows[32]–a plumelet (i.e., upper-mantle plume migration without significant tail and a mushroom head[15])–here.

The lithospheric heterogeneity documented beneath the region[14] can be attributed to Tethyan plate orogeny, as evidenced by numerous observations. The northern branch of the slab subducted beneath Eurasia created arc and back-arc volcanism, mostly dating from the Jurassic to Eocene (e.g., ref. 20 and references therein)[27,33,34]. This part of the slab broke off around the Early Eocene, triggering crustal shortening in the Greater and the Lesser Caucasus by the Oligocene[35,36]. Concurrently, the southern branch (Bitlis-Zagros slab) initiated northward subduction, and the collisional front started to form in response to the Eocene-Oligocene Zagros slab roll-back with the mantle window (slab gap) opening between the Zagros and the Bitlis slabs (ref. 37 and references therein). Subduction jumped northward (~Eocene, e.g., ref. 34), and the closure of the oceanic basin persisted due to the ongoing collision between the Anatolian-Southern Armenian Block and Pontides active margin[26,34]. This phase was alternatively characterized by the initiation of south-dipping subduction of the Caucasus Basin along the arc (e.g., ref. 20). Terminal closure along the Bitlis-Zagros suture (Late Oligocene–Middle Miocene, Fig. 2a) may have led to slab detachment beneath eastern Anatolia at ~25–10 Ma[38,39] (ref. 40 and references therein) Fig. 2b). Ongoing convergence reactivated Pliocene uplift, folding in the Kura Basin with crustal shortening in the southern Greater Caucasus, intensified exhumation across the Greater Caucasus, and strike slip deformation to the south (e.g., North and East Anatolian faults, Figs. 1a, b, and 2c)[16,19,20,27,35,36,41]. Consistent with numerous studies, such as river network analyses[41], the initial break-off of the southern Neotethys slab beneath the Bitlis-Zagros suture zone could have triggered mantle upwelling, propagating from SW to NE beneath the Arabian Plate - a plumelet (Uluocak et al.[15], Fig. 2d). Among various slab deformation models, removal mechanisms involving spatiotemporal overlapping processes[15] can be primarily characterized by: i) wholesale or progressive Bitlis-Zagros slab delamination along the Arabian collisional front in the south (ref. 1 and references therein)[38]; and ii) delamination beneath the Transcaucasus (e.g., ref. 26 and references therein) accompanied by northward plate tearing toward the Greater Caucasus (e.g., ref. 21). Although 3D heterogeneities beneath the region are still debated, magmatic records provide evidence of mantle-crust interactions as a consequence of a discontinuous/semi-episodic closure of the Neotethys Ocean (e.g., ref. 34).

Post-collisional volcanism initiated (around the early–mid-Miocene) on the Erzurum-Kars Plateau (locations shown by T in Fig. 1d) and migrated south and eastward, towards the Bitlis-Zagros Suture and the Armenian border, respectively, consistent with radiometric age estimates (e.g., ref. 42, age compilation in ref. 1). Late Miocene-Quaternary calc-alkaline to alkaline magmatism is documented across the collisional front with ages decreasing westward and subduction signatures weakening southward, towards the Bitlis-Zagros suture (ref. 38 and references therein). Based on chemical and petrological analyses of widespread alkaline lavas active since the Late Cenozoic in the Turkish-Iranian Plateau, it has been argued that the volcanism may be linked to a low seismic velocity zone/body extending to depths of ~300 km[43]. The western and central Greater Caucasus host extensive Neogene–Quaternary magmatism (e.g., Elbrus, Kazbek[16,44]), contrasting with the eastern segment, and highlighting potential along-strike variations in mantle dynamics. Quaternary volcanism in the Lesser Caucasus (e.g., including Kazbek and Georgian volcanic rocks with varying degrees of subduction components[45]) shows compositional variability and distinct melting conditions, indicating small-scale convective upwellings and lateral contrast in lithospheric depths[17,46]. Yet, the regional-scale interactions of long-wavelength asthenospheric processes–including potential plumelet activity[15] (Fig. 2d)–remain poorly constrained in such settings.

Overall, the Arabia-Eurasia collision domain exhibits a complex interplay of multistage tectonics with diverse volcanism, plateau uplift and active fault systems - all linked to Neotethyan slab dynamics involving advance, retreat, and eventual break-off[27,35–37,44,47] with growing evidence suggesting key contributions from mantle convective flows[14,15].

Here, our comprehensive geology-geophysics-geodynamic analyses may help to illuminate the tectonic anatomy of this complex post-subduction system. By coupling mantle flow models with thermal and isostatic constraints, we reveal not only a SW-NE oriented plumelet from the Arabian foreland injecting into the subduction system[15], but also 3D thermomechanical erosion of the upper plates and the active Neotethys slab deformation along the southern collisional front. Integrated data further expose distinct west-to-east differentiation in deformation patterns across the Greater Caucasus, which modifies tectonics at the northern margin of the collision. Our results establish constraints on drip-like removal beneath the post-orogenic magmatic arc under convective forces. Compared to the well-studied seismotectonics beneath the Greater Caucasus Range, in this work we try to clarify the poorly defined tectonic implications of crustal and subcrustal earthquakes along the Arabian collision front between the Arabian-Anatolian-Iranian plate boundary. Crucially, our data-model integration uncovers a previously unrecognized complexity in post-subduction tectonics, providing a transformative framework for understanding active continental collisions worldwide.

## Results

We determine the regional mantle circulation and lithosphere deformation with a 3D instantaneous thermomechanical model based on seismically defined upper mantle structures (Fig. 3) by following the conventional modeling approach discussed in detailed by Uluocak et al.[14,15] in the Arabian-Eurasian collisional region. The results define the anatomy of the structures and dynamics that interact in a complex way in this active post-subduction system of the Arabian-Eurasian collision zone.

Mantle dynamics in the region are dominated by a focused rapid SW-NE directed inflow from the Arabian foreland to the eastern Greater Caucasus–a plumelet–penetrating through the complex morphology of the post-collision environment (Fig. 3). In the southeast, the upwelling mantle beneath the Urumieh-Dokhtar magmatic belt in the Iranian Plateau migrates along-strike of the Zagros mountains, reaching the stable Scythian Plate. The surface response to convective flows (i.e., dynamic topography, e.g., ref. 48 and references therein) is dominated by a positive dynamic support on topography (up to ~800 m) across the Turkish-Georgian-Armenian volcanic zone,

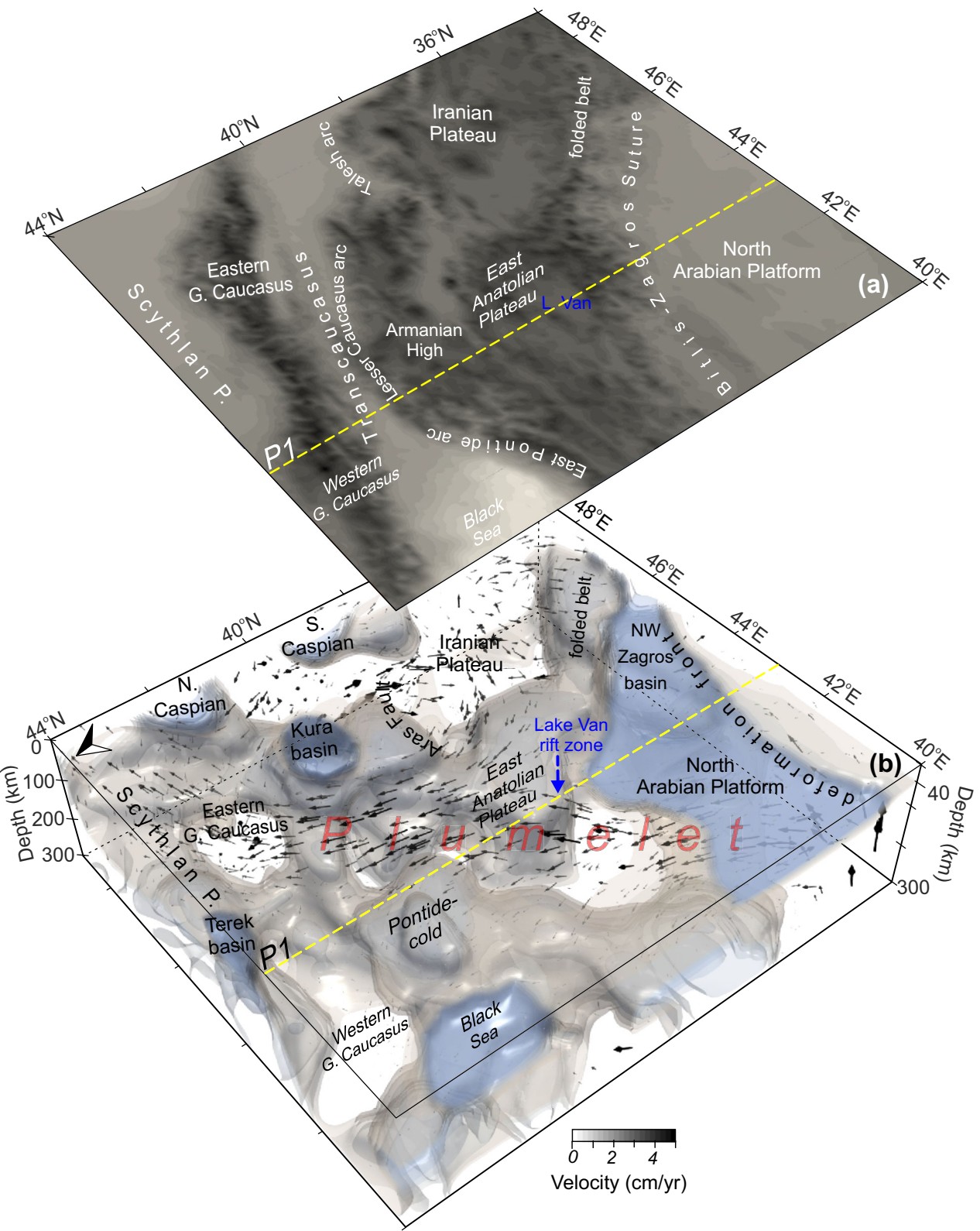

**Fig. 3 | The 3D temperature and vector fields for the depth range of 40–300 km.**
**a** Profile -P1- and geographic locations are marked on topographic relief. **b** Velocity vectors show prominent SW-NE flows – plumelet[15] – from the North Arabian Platform to the East Greater Caucasus (see profile -P1- in Fig. 5). Iso-surfaces are imaged only for ≤1000 K (see color scale in Fig. 4d). MATLAB-R2024a, ParaView and CorelDraw2025 were utilized for the visualization of results.

East Anatolian Plateau, including Lake Van rift province, North Iranian Plateau and the western Greater Caucasus (Fig. 4a). Negative dynamic topography (down to −1.6 km) occurs in basins, e.g., the Kura, Terek, western Caspian, and eastern Black Sea. In contrast to highlands (e.g., the East Anatolian and the Armenian plateaus), basins with underlying cold mantle lithosphere (e.g., blue zones with ≤1000 K; Fig. 3b) and low upper mantle seismic wave attenuations[30] show a similar high shear stress pattern (>10 MPa) in the modeling results (Fig. 4b). The 3D flow model reveals lithospheric-scale zones of prolonged high strain rates (e.g., >2 × 10$^{-15}$ 1/s at the depth of 100 km; Fig. 4c), delineating a SW–NE trending deformation belt along the northern Lake Van intraplate rift zone and the SE-NW oriented Sanandaj-Sirjan and Zagros fold-thrust belts (Fig. 3c). High strain rate anomalies also emerge beneath the North Arabian lithosphere, particularly surrounding the deep NW Zagros, western Caspian, and Kura basins (Fig. 4c).

The flow modeling indicates regional deformation zones in the central and eastern Greater Caucasus extending to the stable Scythian Plate, marked by visible lateral contrast of low shear stresses in the crust (≤1 MPa; Fig. 4b) with high strain rates beneath the crust (Fig. 4c). The low shear stress and strain rates with the positive dynamic topography manifest in the TGA volcanic highlands and the West Greater Caucasus is in concordance with hot upper mantle anomalies (≥1500 K; Fig. 4d).

To elucidate mantle controls on crustal dynamics, we integrate seismically-derived thermal state with crustal isostasy analysis using various independent datasets (Fig. 5). We show P- and S-wave velocity ratios, which can give insights into melting conditions in the case of upwelling mantle at 100 km depth and for the crust in the region (e.g., refs. 49,50). The highest subcrustal velocity ratios (≥1.9) occur along magmatic arcs. Slightly lower values (~1.8–1.9; Fig. 5a) are observed in the North Arabian Platform and the East Anatolian Plateau. In terms of crustal velocity ratios (green contour; Fig. 5a), compared to global crustal averages of ≥1.77[42], the highest average values are spatially correlated with the highest mantle velocity ratios in magmatic arcs and the West Greater Caucasus, with an interesting exception of the Central Greater Caucasus region. This demonstrates a direct mantle-crust linkage in concordance with crustal and subcrustal thermal anomalies beneath young volcanic centers (Fig. 5a) and our 3D temperature model (Figs. 3b and 4d).

We also provide a comparison of regional velocity ratios, crustal thickness, and topography, along with overall data from other areas (Fig. 5b,[51]). Although there are local differences arising from different data sets, the changes in the South Caspian, Zagros, and Black Sea basins and dynamically uplifted continents can be identified by cold (i.e., yellow zones with lower elevation) and hot features (i.e., red zones with ~40–45 km crustal thickness), respectively, in Fig. 5b. Vp/Vs values in the Armenian Plateau, for instance, fall in a zone (A; Fig. 5b) that is classified by a crustal thickness of ≤-36 km[42], yet has a high elevation (≥1.5 km). These results agree well with positive dynamic topography, estimated by our thermomechanical model (Figs. 4a and 6a).

Consistent with the hot crustal thermal regime (Fig. 5), regional mantle structures are defined by relatively shallow, arc-parallel relatively cold anomalies (<1500 K) beneath the Transcaucasian zone (including the eastern Pontides to the Lesser Caucasus, Fig. 4d). Our temperature model reveals a distinct Pontide-cold anomaly at shallow mantle depths (~80–250 km; Figs. 4d and 6b) that exhibits two key characteristics: i) it is decoupled from the overlying crust beneath the volcanic plateau (Dzhavakheti Province of South Georgia); and ii) it is spatially correlated with the surface expression of young volcanoes and the adjacent Armenian magmatic highland (Figs. 3a and 5a).

In the active Arabian collisional front, the results reveal an interaction between the plumelet and cold upper mantle structures, namely a mantle anomaly beneath the thick northern Arabian lithosphere (~250 km depths; Fig. 6b). This anomaly differs from the NW Zagros depression area (P2; Fig. 6d, e) in that it is decoupled from the upper

plate, implying fully detached cold material- termed as a part of a detached slab or delaminated Bitlis fragments- sinking beneath the North Arabian Platform. Deep basins, such as Zagros and Kura basins (Fig. 6d, e) are characterized by their cold roots and negative dynamic support on the surface. We define an active slab breaking off beneath the East Greater Caucasus with a cold thick root of stable Scythian Plate at the northern boundary of the collision (P2 in Fig. 6e).

The temperature model (Fig. 6e) depicts the southeast subducted, actively tearing Zagros Slab along the southern collisional margin (Fig. 7c). We define the fully and partially detached Bitlis slab in detail, along a cross-section in Fig. 6 and as a 3D temperature model in Fig. 7. Notably, our findings provide compelling evidence for plumelet-induced thermomechanical erosion with the 3D mantle thermal structure (Fig. 7b) revealing a distinctive spatial signature within the detached/sinking Bitlis Slab. The results demonstrate the regional thermomechanical erosion beneath the crust (~40 km) of the Arabian foreland in the form of a plumelet-fed low-viscosity zone (red star in Fig. 7).

## Discussion

The calculated mantle flow vectors align closely with observations, such as modern kinematics (Fig. 1a), seismic anisotropy orientations, and a low seismic velocity zone (Fig. 1c) that extends to the East African plume-magmatic belt at large-scales[31,32]. These results reconcile previously disparate observations - from shallow (<300 km), fast seismic velocities (at ~250 km depth[24]) to overlying slow seismic anomalies (e.g., refs. 24,31; the plumelet[15]) into a unified geodynamic framework validated by different numerical models[14] based on recent seismic tomography data[29].

Most notably, our results help explain distinct patterns of seismicity across the collisional front. Subcrustal earthquakes (blue circles; Fig. 7c, d) are tightly clustered within the horizontally deforming upper plate, precisely mapping the active tear zone, while the plumelet-fed thermally eroded region to the west (red star; Fig. 7a−d) remains aseismic. This contrast highlights the fundamental difference in deformation mechanisms in the south, between east and west parts of the subducted zone (e.g., Fig. 7d). Furthermore, crustal seismicity (yellow dashed line; Figs. 1 and 7) is strongly concentrated along the plate boundary, where both our numerical models (Fig. 6) and geophysical observations (e.g., gravity data[52]) reveal pronounced lateral contrasts in uppermost mantle deformation patterns. While future tectonic rupture analyses could provide additional resolution, these findings already reveal how mantle processes control the spatial partitioning of seismicity.

The southeastward propagation of plate deformation is modulated by the plumelet and the pull[52] of the Zagros Slab (Fig. 6e, e.g., refs. 53,54) as evidenced by the high lateral contrast in temperature, stress, strain rates (Figs. 4 and 6e), and viscosity variations (Fig. 7c). This process explains both the rapid NW Zagros Basin subsidence (white stars; Figs. 4c and 7a) and the intense mantle-sourced magmatism during the middle-late Miocene (e.g., ref. 52), as well as the fragmented high-viscous materials shown in the flow models (Fig. 7c).

The results reveal contrasting deformation regimes in the Greater Caucasus, with distinct western and eastern domains evidenced by differential exhumation and convergence rates across the high mountain range[55]. In the central-eastern sector (Figs. 4 and 6d, e), we attribute these variations to an active slab break-off process, consistent with well-documented tectonic stress accumulation (Fig. 1b) and northeastward tearing at the plate margin (e.g., refs. 19–21, [25] and references therein). In the western Greater Caucasus, the crust-mantle interaction coincides with positive dynamic topography (Fig. 4a), heating from the mantle, low-to-ultralow seismic velocities, elevated surface heat flow, and geothermal gradients[14,19,24]. Geochemical analyses of Neogene-Quaternary magmatism with locations of the Holocene volcanoes in the western Greater Caucasus and the western edge

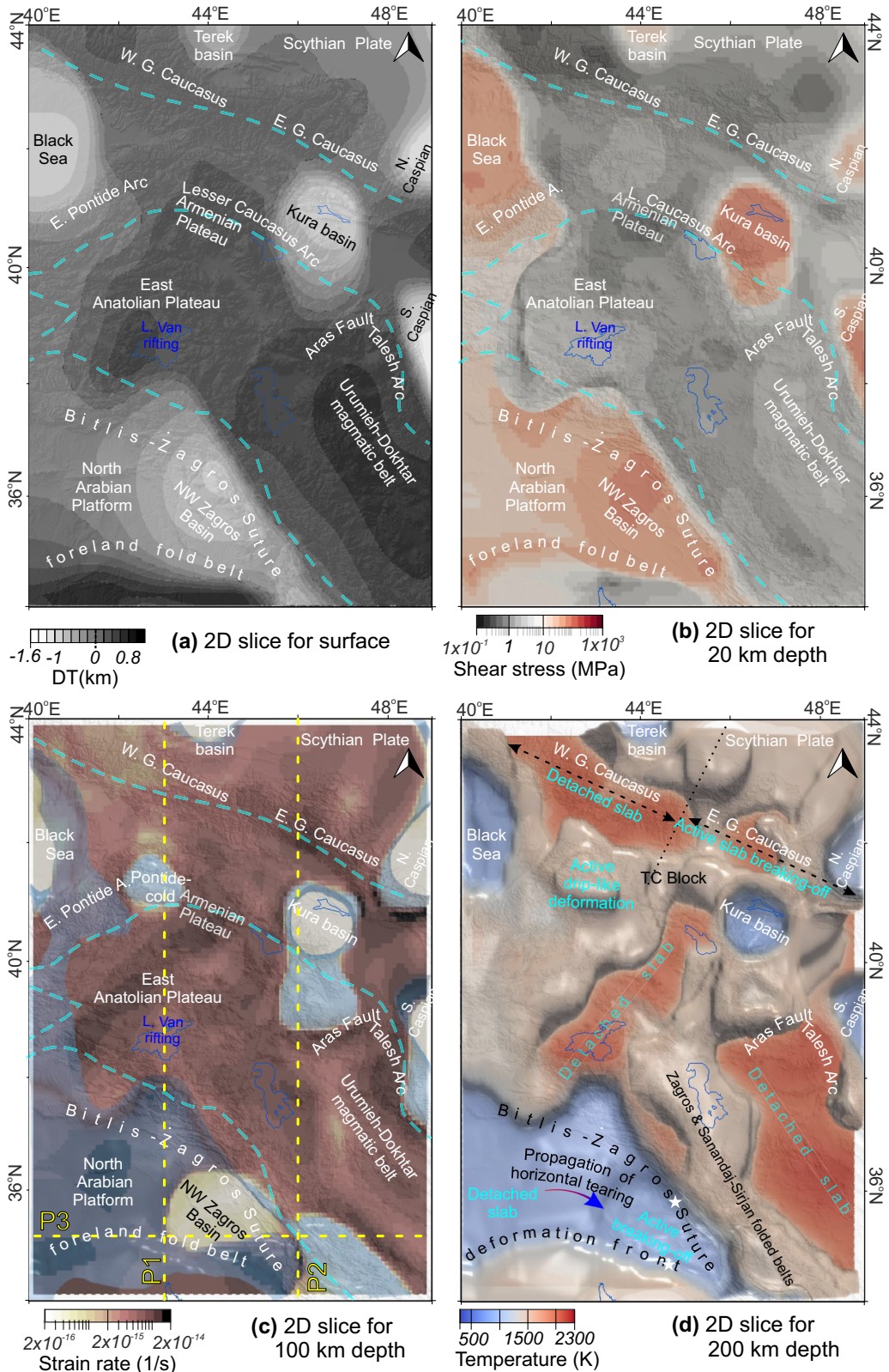

**Fig. 4 | Regional deformation patterns from surface to subcrustal depths based on 3D instantaneous numerical modeling results. a** Upper mantle-induced surface deflection (dynamic topography; DT). **b** Amplitudes of the shear stress in a slice at 20 km depth. **c** Strain rates at the depth of 100 km. Blue iso-contours in (**c**) show cold (1300 K) anomaly for a depth range of [40, 100] km. Dashed yellow lines show the profile 1-P1. **d** The 3D temperature model for a depth range of [40–200] km with a temperature slice at the depth of 200 km. White stars show the proposed location of the NW Zagros flexural basin subsidence in (**d**) and Fig. 7a (see the text for references).

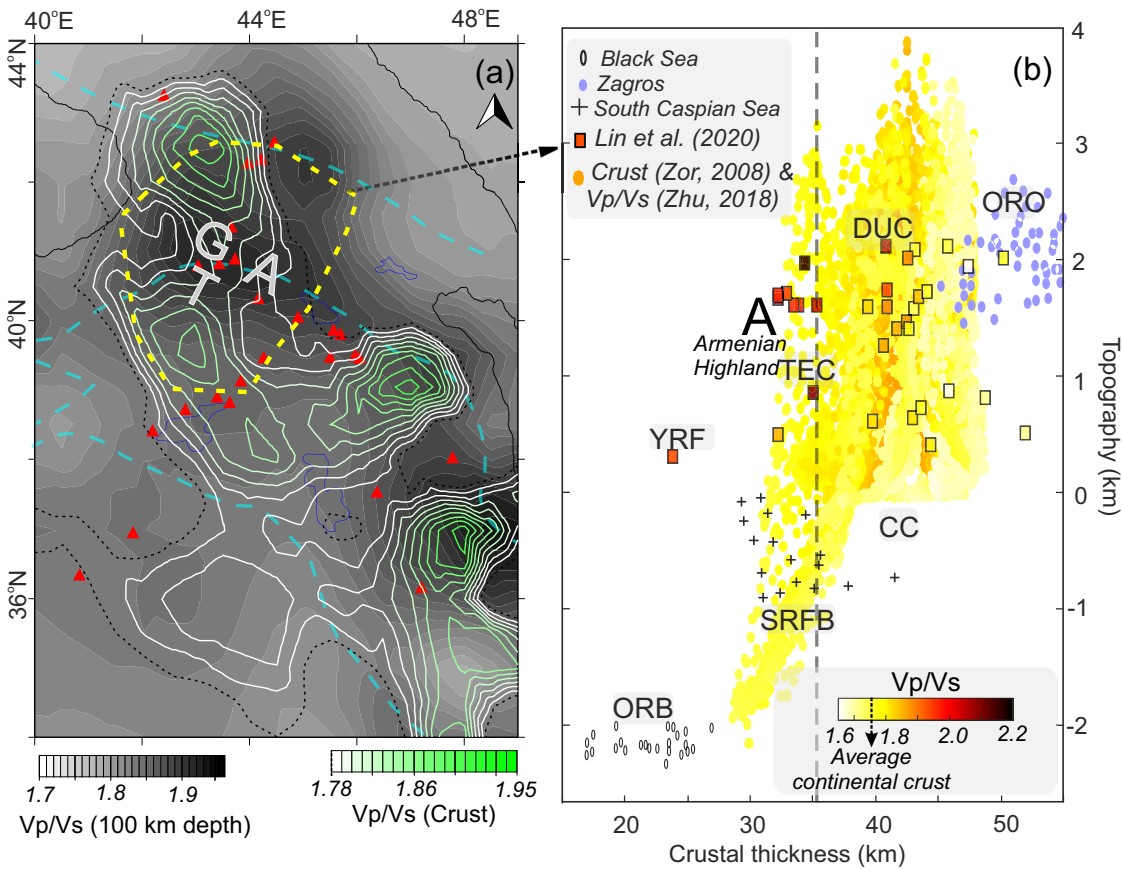

**Fig. 5 | Thermal and isostatic states of the region. a** Average of Vp/Vs ratios for the crust (at 20 km and 40 km depths) and Vp/Vs ratios for the depth of 100 km (modified from refs. 49,50). Variations higher than the average continental crust (ref. 42 and references therein) are shown as a contour map (green lines). **b** Crustal thickness vs. topography variations. The selected area (dashed yellow line) in (**a**) indicates high-resolution crustal Vp/Vs observations[42,78] also shown as rectangular symbols in (**b**). The letter A stands for the Armenian volcanic highland with

relatively thin crustal thickness (gray dashed line) and is interpreted as concerning partially melted lower crustal features[42]. (See Fig. 1 further symbols and boundaries). CC Cratonic Continent, DUC Dynamically Uplifted Continent, ORB Old Rifted Basin, ORO Orogens, SRFB Subducted-Related Foreland/Forearc Basin, TEC Thermally Eroded Continent, YRF Young Rifted Basin[51], TGA Turkish-Georgian-Armenian plateau.

of the actively deformed central Greater Caucasus (e.g., Kazbek; Fig. 1a) support our numerical results and analyses. Notably, the mismatch between crustal and lithospheric-scale velocity ratios in the central Greater Caucasus (Fig. 5a) suggests an incomplete mantle window (Fig. 4d) that may facilitate crustal heating during eastward propagation of plate rupture adjacent to the detached slab zone, i.e., the western Greater Caucasus.

Our integrated analysis (Fig. 5) demonstrates a systematic correlation between high average Vp/Vs ratios (≥1.8) at elevated regions (≥1 km) and relatively thin crusts (≤40 km), indicating upper mantle-driven non-isostatic support. The Turkish-Georgian-Armenian (TGA) volcanic provinces, East Anatolian Plateau, western Greater Caucasus, Talesh (Azerbaijan), and Urumieh-Dokhtar magmatic zone are defined by isostatically under-compensated topography with thermally heated crustal and uppermost mantle structures. Conversely, negative dynamic support, associated with over-compensated topography and thick, dense crustal roots (basins, Fig. 4d), corresponds to low uppermost mantle Vp/Vs velocity ratios. These analyses are consistent with Airy-type isostasy models[13–15] and support Zhu's (2018) interpretation of anomalously high Vp/Vs ratios accompanied by slow S-wave velocities, reflecting partial melting beneath the Anatolian Plate and its surroundings. Regional studies[20,42,56] corroborate our results, particularly in areas experiencing mantle-derived heating and post-collisional magmatism, such as southern Armenia with partially molten lower crustal features.

We identify small-scale cold anomalies that are also visible in independent seismic tomography models beneath the Eastern Pontides-Lesser Caucasus arcs and the East Anatolian Plateau[14,23,54]. These relatively shallow, seismically fast bodies are typically interpreted as either remnants of the Tethyan oceanic lithosphere (e.g., the TC block in Fig. 3d[57]) or as buoyant/floating fragments of the delaminated Neotethys oceanic slab (e.g., refs. 19,46,54), which have remained beneath the collisional zone since the middle-late Miocene. Based on our results, the Pontide-cold indicates a drip-like removal under the southern Georgian volcanic province by significant influence of plumelet-induced convective forces. Further, the subduction-modified mantle composition observed in the magmatic arc of the Transcaucasia can be attributed to the plumelet-induced drip deformation, in addition to magmatism associated with: (1) prior (i.e., in relation to inherited water and slab fragments from previous subduction events [ref. 17 and references therein]; or (2) coeval with the slab break-off (wholesale break-off or gradual delamination of the southern branch of the Neotethys slab[38, 1] and references therein). During lithospheric dripping, both long-wavelength – plumelet, as shown here, and/or plume-induced small-scale convection cells suggested by numerical models[58] and short-wavelength flows[59] – can contribute to small-scale convective removal and partial melting not only beneath the detached zone (Fig. 5a) but also the neighboring area. This interpretation is corroborated by relatively local crustal thinning, thermal heating, and magmatism with mixed geochemical signatures

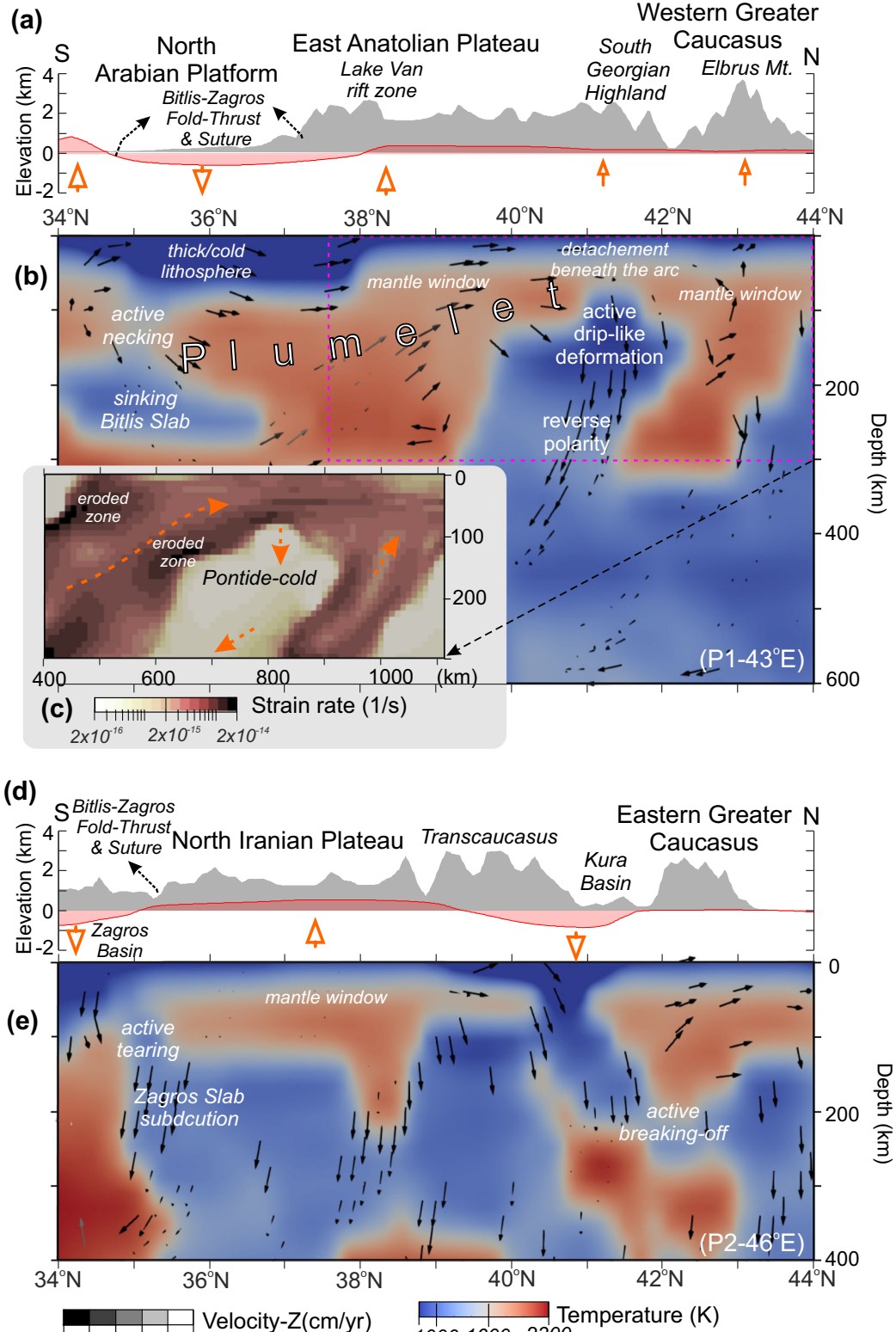

**Fig. 6 | Spatial variations of drip-like arc deformation and slab foundering at plate margins.** Dynamic (red) and observed topography (gray) along profile P1 (**a**) and P2 (**d**). Temperature model with superimposed convection vectors along P1 (**b**) and P2 (**e**) (vector colors scaled by vertical velocity components). **c** Strain rate distribution within the selected area in (**b**). The Pontide-cold anomaly exhibits low temperatures and reduced strain rates, accompanied by downwelling flow (orange vectors indicate direction of motion).

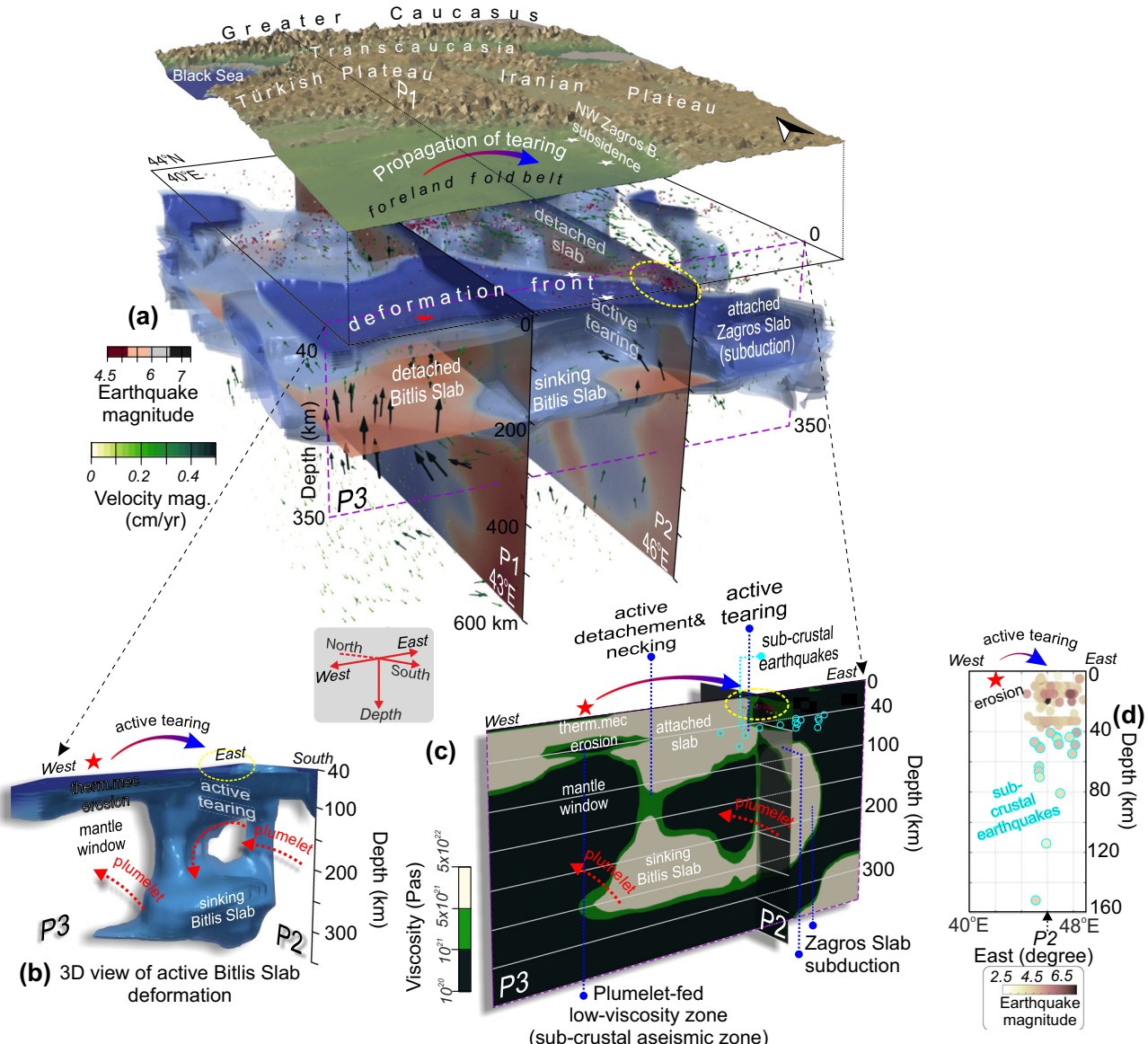

**Fig. 7 | Neotethys slab foundering mechanisms at the Arabian-Turkish-Iranian convergence margin with different slab deformation patterns. a** Temperature field (iso-contours for [40–300] km depth range) with superimposed velocity vectors. Earthquake distributions (dots; see Fig. 1a) indicate active seismotectonic deformation. The interaction between the convective forces and the actively necking, tearing and breaking Bitlis and Zagros slabs beneath the convergent margin. **b** 3D perspective of the cold, sinking Bitlis Slab. **c** Viscosity variations along profile P3 (and the south part of P2) with different slab foundering mechanisms. Dashed red arrows indicate mantle flow directions, red stars mark the plumelet-induced low-viscosity zone. Yellow ellipse and blue circles denote hypocenters of crustal and subcrustal earthquakes, respectively. **d** The longitudinal distribution of earthquakes between 34°N-35°N (ref. 79, between 1900 and 2025). Red stars indicate thermomechanically eroded zones.

documented in the Armenian volcanic highland[45,46,57]. Although region-specific geodynamic models (e.g., ref. 6) provide additional kinematic constraints, our drip hypothesis establishes a useful approach for interpreting both a plumelet and Pontide-cold anomaly. We therefore interpret that there is a Pontide-cold-related magmatic arc cycle here supported by observational (geological-magmatic-tectonic) signatures changing from arc to an intraplate deformation above former subduction zones within continental collision systems (e.g., refs. 60–63). Our findings also demonstrate a flow polarity reversal at the base of the detached drip material (southward flows; Fig. 6c), which may be caused by the weak arc features[63]. Present-day deformation patterns from our numerical model and documented fluctuations in uplift rates along the Arabian-Eurasia collision zone since ~5 Ma[15,41] indicate non-steady-state mantle dynamics. This dynamic instability challenges conventional models for collisional orogeny and highlights the

importance of regional deformation mechanisms-plumelet interactions in forming post-subduction tectonics.

A key finding of the work is in showing how pronounced contrasts in surface observables can be used in determining plate boundaries, which may play important roles in collisional zones, such as the Aras Fault hosting the long-lived Cenozoic igneous activities between eastern Caucasus and western Iranian segments[37]. The surface trace of lithospheric-scale discontinuities might be difficult to define from direct observations in collisional settings, especially in zones covered by region-wide highly deformed volcano-sedimentary units (e.g., ref. 64). However, our thermomechanical model reveals deep-seated inherited plate boundaries as guiding the interplate deformation pattern between converging blocks (Fig. 4).

Our overall results demonstrate that a plumelet injecting from the southwest drives major regional deformation along its propagation

channel, modifying the morphology of the subducted lithospheric slabs of the post-subduction Neotethyan collision (Fig. 3). This documents a first-order control of mantle dynamics on plate rupture mechanisms at tectonic margins. The resulting deformation architecture is expressed by short- and long-wavelength dynamic topography (volcanic highlands, including the intraplate Lake Van rift zone and deep basins; Figs. 4a and 6a, d), a deep-seated deformational front (foreland fold belt; Fig. 7a) along the lithospheric channel from the Arabian foreland to the eastern Greater Caucasus, and thermomechanical erosion (Fig. 7, red star in Fig. 1c) −collectively providing direct evidence of plumelet-induced processes on continental collision tectonics. Further, the results explain the distinct spatial patterns of seismicity across the Arabian collisional front, linking plate tearing to crustal and subcrustal earthquake distributions through 3D deformation patterns.

In the tectonic anatomy of the complex plate boundary, during its migration from deep mantle source(s) in the south, the plumelet reworks the subducted lithosphere actively converging with Eurasia. That is, the plumelet plays an active role in plate consumption, as demonstrated by thermomechanical erosion in the overlying plate and the sinking slab along the active collision margin, and by drip-type removal beneath the volcanic arc. The results reveal a more intricate multi-scale behavior of the post-collisional tectonic boundary than the traditional models (e.g., refs. 1,38) based solely on slab dynamics and boundary forces. Such a complex anatomy may be applicable to other complex zones of collision, such as in Atlas (e.g., ref. 65) or in the Arabian Peninsula (blue star; Fig. 1c), where mantle material−plumelets−are able to propagate laterally through the fragmented/broken post-subduction plate boundary. Or perhaps such mantle flow is a ubiquitous signature of post-collisional environments that we are just now able to resolve. The findings contribute to an enhanced understanding of the evolution of tectonic collisions and the mantle forces driving them.

## Method

We employed state-of-the-art numerical modeling using a well-established framework applied to diverse tectonic settings (e.g., Central Anatolia, SE Carpathians; ref. 15 and references therein). We used seismic data[12] to constrain the geometry of lithospheric structures and mantle features for the model[15,66]. Since estimations of instantaneous thermomechanical models are data-dependent, sensitivity analyses for the modeling approach used here were performed and documented in previous work on the collisional zone[14,15]. Our model leverages ASPECT (ref. 67 and references therein) to solve the governing equations of the conservation of mass:

$$\nabla . \mathbf{v} = 0 \tag{1}$$

where $\mathbf{v}$ is the velocity (m/s), momentum:

$$\nabla . \sigma_{ij} + \rho g = 0 \tag{2}$$

$\sigma_{ij}$ is the stress tensor; g is gravitational acceleration (9.8 m/s$^2$), $\rho$ is density (kg/m$^3$) and energy:

$$\rho C_p \left( \frac{\partial T}{\partial t} + \mathbf{v} . \nabla T \right) = k \nabla^2 T + \rho H \tag{3}$$

$C_p$ is specific heat capacity (J/kg/K), T is the temperature (K), t is time(s), k is thermal conductivity (W/m/K), H is the rate of internal heat production per unit mass (W/m$^3$) incorporating the visco-plastic rheology for an incompressible medium (ref. 67 and references therein). The effective viscosity ($\eta_{eff}$) in the viscous stress is defined for

**Table 1 | Parameters used in the experiment (ref. 14 and references therein)[69,70]**

| Mechanical parameters | Crust (wet quartz) | Mantle lithosphere (dry olivine) | Asthenospheric mantle (dry olivine) |
|---|---|---|---|
| Density (kg m$^{-3}$) $\rho_0$ | 2840 | 3300 | 3260 |
| Viscosity pre-factor (B; Pa$^{-n}$ s$^{-1}$) | $8.57 \times 10^{-28}$ | $6.52 \times 10^{-16}$ | $6.52 \times 10^{-16}$ |
| Power law (stress) exponent ($n$) | 4 | 3.5 | 3.5 |
| Activation energy (Q; kJ mol$^{-1}$) | 223 | 530 | 530 |
| Thermal conductivity (k; W m$^{-1}$ K$^{-1}$) | 2.5 | 2.25 | 2.25 |

The internal friction angle ($\varphi$) = 20° and the specific heat capacity ($C_p$) = 750 (J kg$^{-1}$ K$^{-1}$) for all materials. Activation volumes (V) = $18 \times 10^{-6}$ (m$^3$ mol$^{-1}$) for the materials beneath the crust. The reference strain rate = $1 \times 10^{-14}$ s$^{-1}$, the viscosity limits are $\eta_e = 1 \times 10^{20} - 5 \times 10^{22}$ Pa s (e.g., ref. 14 and references therein). The square root of the second invariant of the strain rate tensor is shown as strain rates (1/s) for simplicity in the text.

the power law creep based on the temperature and the square root of the second invariant of the strain rate tensor ($\dot{\varepsilon}$);

$$\eta_{eff}(\dot{\varepsilon}, T) = \frac{1}{2} \beta B^{\frac{-1}{n}} \dot{\varepsilon}^{\frac{(1-n)}{n}} \exp\left(\frac{Q + PV}{nRT}\right) \tag{4}$$

where n is the non-Newtonian viscosity exponent, Q is the activation energy, V is activation volume, R is the universal gas constant (8.31 J/mol/K), B is the viscosity pre-factor and $\beta$ is the scaling factor, P is the total pressure[67] (Table 1). The deviatoric stress; $\sigma_{ij} = \min\{\sigma_y; \sigma_v\}$ in relation to the deformation of the material was defined based on the plastic yield stress ($\sigma_y = P \sin\varphi + C \cos\varphi$, i.e., Drucker−Prager yield criterion for the brittle failure, where $\varphi$ is the internal angle of friction, C is cohesion) and the shear stress tensor ($\sigma_v = 2\eta_{eff}\dot{\varepsilon}$). The temperature model (e.g., Figs. 3b, 4d and 5e) was obtained by using seismic data with a density scaling approach based on the thermal-expansion expression; $\rho(T) = \rho_0 (1 - \alpha \Delta T)$ (reference densities, $\rho_0$, are listed in Table 1; a coefficient of thermal expansivity, $\alpha$, $2 \times 10^{-5}$ 1/K was used) with the sampling intervals of 20 km in each direction for the investigated region[15]. The time stepping in the model was carried out using a small Courant−Friedrichs−Lewy number of 0.1, yielding a timestep of up to ~$1 \times 10^5$ years and the relatively small-velocity errors (i.e., RMS ≤2 cm/year) with the effective computation time[14]. Dynamic support in the topography- dynamic topography (e.g., ref. 13)- is defined along the free surface based on the vertical stress exerted on the lithosphere as a response to normal fluid stresses in the underlying mantle. Any possible responses of elasticity, mass transfer, and radioactive heating were not considered in the incompressible solution domain. Since different numerical models with various temperature configurations and rheological features were tested[14,15], in this study we followed a convection modeling approach with inputs that have already been confirmed/validated by considering large sets of geophysical data and analyses documented in the Arabian-Eurasian collisional region. For a more detailed discussion of the modeling methodology and parameters, the reader should refer to refs. 15,67,68.

## Data availability

Model inputs used in this study have been deposited in the Uluocak (2024a) under [https://doi.org/10.5281/zenodo.12568287]. Seismic data used in this study were adapted from Piromallo & Morelli (2003) that is available under [https://doi.org/10.1029/2002JB001757].

## Code availability

Open-sourced ASPECT codes used in the study are available under [https://zenodo.org/records/6903424].

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

## Acknowledgements

We acknowledge the Computational Infrastructure for Geodynamics (geodynamics.org), which is funded by the National Science Foundation under award EAR-0949446 and EAR-1550901 for supporting the development of ASPECT (ver. 2.4, https://aspect.geodynamics.org/). Computations were performed on the GPC supercomputer at the Niagara HPC Consortium with support provided by the Department of Earth Sciences, University of Toronto (www.es.utoronto.ca) and the Digital Research Alliance of Canada (www.alliancecan.ca). E.Ş.U. sincerely thanks to Claudia Piromallo for her discussion on validation of the

P-wave seismic tomography model by Piromallo & Morelli (2003). E.Ş.U. also thanks Sascha Brune for his general discussions on numerical modeling and Şeref Uluocak for valuable perspectives on the philosophy of modeling. We acknowledge the use of ParaView (https://www.paraview.org), GeoMapp (http://www.geomapapp.org), CorelDraw 2025 (https://www.coreldraw.com) and MATLAB-R2024a (Version: 24.1.0.2578822.https://www.mathworks.com) for the visualization of data and results. E.Ş.U. acknowledges TÜBITAK (BIDEB-2219) for support by the International Postdoctoral Research Fellowship Program. This work has been supported by TÜBITAK (BIDEB-2219) International Postdoctoral Research Fellowship awarded to Ebru Şengül Uluocak. Numerical model has been made by financial support from the Department of Earth Sciences, University of Toronto (https://www.es.utoronto.ca) and Compute Canada (https://www.computecanada.ca).

## Author contributions

E.Ş.U. conducted conceptualization, formal analysis, investigation, methodology, modeling, visualization, supervision, writing original draft, discussion, editing and revising the manuscript. R.N.P. provided computational resources for numerical modeling, conducted discussion, conceptualization, writing original draft, review and editing. C.F. contributed to the discussion, editing and revising the manuscript. T.S. contributed to the discussion, editing and revising the manuscript.

## Funding

## Competing interests

The authors declare no competing interests.
