## [Transparent Peer Review file · Nature Communications]

Anatomy of a post-subduction collision

Corresponding Author: Dr Ebru Şengül Uluocak

**This file contains all reviewer reports in order by version, followed by all author rebuttals in order by version.
Redactions – published data**

Version 0:

Reviewer comments:

Reviewer #1

(Remarks to the Author)

I read this manuscript with great interest as it presents a novel perspective on the Arabia-Eurasia collision zone, drawing upon a synthesis of geological and geophysical data, as well as thermo-mechanical modelling. This new perspective is supported by the data, and I recommend that it be considered for publication in Nature Communications. However, I believe that some points still need to be clarified before it can be accepted.

Firstly, it seems that the authors are pitting their model against a 'traditional model', but the latter is not clearly explained in the present paper. In my view, there are several theories regarding the evolution of the Caucasus region. Some previous papers suggest a significant role for delamination (cf. Grosjean et al., 2022), a process which is not clearly discussed or taken into account in this paper. Some papers have already proposed the contribution of the deep asthenosphere to the magmatism of the area (e.g. Soltanmohammadi et al., 2018), thus excluding any purely 'collisional' component. Previous models did not strictly fall into a single category, and I agree that the horizontal mantle flow or 'plumelet' model proposed here is novel, as previous models assumed that the mantle flowed mainly vertically.

On the other hand, in the 'Results' section, the authors immediately propose a model without first presenting the results in a cautious or analytical manner. For example, at lines 197–199, they claim that 'Mantle dynamics are dominated by a focused, rapid, SW–NE-directed inflow from the Arabian foreland to the eastern Greater Caucasus—a plumelet—penetrating through the complex morphology of the post-collision environment'. While I believe this interpretation is entirely plausible, it is crucial to present results more factually and demonstrate how they support the plumelet model.

Further points should be clarified:

What role do vertical structures on the lithosphere scale play in propagating the plume? The inception of horizontal flow within a structure featuring vertical slabs requires zones of weakness inherited from the subduction-early collision history (step faults etc).

- The so-called 'Bitlis slab' may also be the result of delamination, because the Bitlis subduction zone is characterised by HP metamorphism at 74–71 Ma (e.g. Rolland et al., 2012), which likely resulted in sinking at very deep levels of the mantle (deeper than 300 km) and should have been N-dipping (no volcanic arc developed towards the south). This tectonic zone is generally considered to be complex and composed of numerous tectonic blocks and sutures. Therefore, the 'closure of Neotethys' should not be considered a progressive process, but rather a discontinuous 'step-by-step' process in which basins were successively closed. This explains why the timing of 'collision' is disputed in this region.

Overall, the text is well written, but some sections could be improved (I have underlined some examples in the text). Please avoid self-satisfaction (e.g. 'Our comprehensive geological, geophysical and geodynamic analyses illuminate the tectonic anatomy...').

The figures are well drafted and informative. However, when dealing with seismicity (Fig. 6), it is difficult to understand the clustering of seismicity. It might be better to present a representative vertical section showing the seismicity in relation to the crustal levels.

I also think the title does not fully reflect the content, as it should emphasise the 'plumelet' hypothesis.

I hope these comments are useful. With best regards.

refs

Grosjean, M., Moritz, R., Rezeau, H., Hovakimyan, S., Ulianov, A., Chiaradia, M., & Melkonyan, R. (2022). Arabia-Eurasia convergence and collision control on Cenozoic juvenile K-rich magmatism in the South Armenian block, Lesser Caucasus. *Earth-Science Reviews*, 226, 103949.

Soltanmohammadi, A., Grégoire, M., Rabinowicz, M., Gerbault, M., Ceuleneer, G., Rahgoshay, M., ... & Benoit, M. (2018). Transport of volatile-rich melt from the mantle transition zone via compaction pockets: implications for mantle metasomatism and the origin of alkaline lavas in the Turkish–Iranian plateau. *Journal of Petrology*, 59(12), 2273-2310.

Rolland, Y., Perincek, D., Kaymakci, N., Sosson, M., Barrier, E., & Avagyan, A. (2012). Evidence for ~ 80–75 Ma subduction jump during Anatolide–Tauride–Armenian block accretion and ~ 48 Ma Arabia–Eurasia collision in lesser caucasus–East Anatolia. *Journal of Geodynamics*, 56, 76-85.

Reviewer #2

(Remarks to the Author)

Anatomy of a Post-Subduction Collision

Authors: Ebru Şengül Uluocak, Russell N. Pysklywec, Claudio Faccenna & Taylor Schildgen

This paper addresses the problem of understanding mantle dynamics in continental collision zones where the driving oceanic slab has detached or broken off from the overriding plate. These post-collisional systems are characterized by prolonged crustal and upper mantle deformation that can persist long after the initial convergence and slab consumption has ceased. The Arabian-Eurasian collision zone exemplifies such complexity, where the Neotethyan oceanic slabs have undergone various stages of subduction, break-off, and foundering, leaving behind a fragmented lithospheric architecture. Understanding what drives continued deformation in these post-subduction environments remains a fundamental challenge in tectonics, as traditional plate-driving forces are absent yet significant surface deformation, volcanism, and seismicity persist.

Overview

The manuscript is well-written and organised and it is of significance because it showcases how mantle dynamics strongly influences strain rate, stress, topography, and mantle flow of post-collisional settings. It also demonstrates how tectonic activity post-dates the interruption of subduction due to the effect of mantle dynamics and provides a framework for the integration of geodynamic modelling with geophysical observables.

The approach is based on correlating various geophysical observables - seismic velocities, topography, crustal thickness, and seismicity patterns - with their 3D thermomechanical model results. The authors systematically demonstrate how regions of high V_p/V_s ratios correspond to magmatic arcs and elevated topography, how cold mantle anomalies correlate with thick lithospheric roots and deep basins, and how modeled strain rates align with observed seismicity distributions. It is these (positive) correlations that form the empirical foundation for validating their model predictions and serve as the primary evidence supporting their interpretations.

Another central concept is the "plumelet" which is defined by the authors as the migration of upper-mantle plume material into the region. It may be drawn in by flow in the region or by its own buoyancy. The plumelet is proposed as a flow of hot mantle material from the Arabian region to the eastern Greater Caucasus. They propose that, from their 3D thermomechanical modeling, this lateral mantle flow *drives* regional deformation, influences slab break-off processes, *causes* thermomechanical erosion of overlying plates, and creates the observed dynamic topography patterns across the Arabian-Eurasian collision zone. There are a number of other areas where subduction influences mantle flow - for example the paper by Miller & Becker (<https://doi.org/10.1130/G34959.1>) argues that uplift of the Atlas is enhanced by flow of plume material around the foundering Alboran slab. Or indeed the paper by Lanari et al (2023) which has some author overlap with this paper. Are the other notable examples that would help to establish this plumelet hypothesis and generalise it ?

A fundamental question raised by this interpretation is whether the plumelets **are** actively driving the observed tectonic signals, or whether they represent more passive structures that are being drawn into the region by the process of slab breaking off and foundering. The distinction between active versus passive roles has significant implications for understanding the causal relationships in post-collisional tectonics and the broader applicability of the plumelet concept to other collision zones. Is the plumelet genuinely causing these effects ? Can it be determined convincingly through the models ?

The Tectonic context and motivation section is well-written, but is hard to follow due to the absence of a figure that provides context for the tectonic evolution of the study area. In Figure 1, the main magmatic units are shown; however, it would be beneficial to provide information about the composition of these volcanic units. Therefore, a figure showing the tectonic evolution, the location of magmatic units and their chemistry is needed. This is very important for readers who are not familiar with the geology of the study area.

In general, figures are very pretty but not always easy to follow. Figure 5 is quite straightforward, but Figure 6a is not. Figure 6b is difficult to see compared to figure 6c. We know the 3D renderings are a lot of work to produce but in static form they are less useful than the cross sections (sorry).

Figure 6 is helpful because it integrates all the observables and the results from the numerical modelling. However, the mantle structure of the study area is quite complex and the 3D geometry of the plumelet propagation channel is not fully clear. We suggest you could draw your interpretation of this geometry in Figure 6b, and this will help the reader to better visualize how it correlates with variations in stress, strain rate, dynamic topography and Vp/Vs ratios.

Detailed Feedback

- Pn and Sn are quite well known but not universally, it would be useful to add a short explanation
- Lines 86-89: In the captions of Figure 1d, are the red triangles showing the magmatic units or active volcanoes?
- Lines 149-150: It is worth mentioning which observations this refer to (geological, geophysical, geodynamic) ?
- Lines 204-205: Figure 2 would benefit from adding a label with the location of the Scythian plate.
- Lines 221-222: differences of less than 1MPa do not seem that "high"
- Line 228: Is there any reason for Figure 2 to display the north at an orientation different from all other figures?. Although you include the labels of the main tectonic domains, it is confusing.
- Lines 249-251: you mentioned "crustal velocity ratios"; however, you didn't mention the crustal depth or depths considered in the contours.
- Lines 253-254: "corroborates" is not the most suitable word for your explanation
- Lines 283-284: What does "Pontide cold" mean?
- Lines 333-336: The vectors in Figure 6 can be improved. This would facilitate the comparison with the observed GPS velocity field. Also, Figure 6 shows a zoomed region, but the extent of this zoomed region is not clear. It would be useful to include an inset with a regional map showing the location of this region.
- Lines 274-275: What is the purpose of the vertical line at ~35 km of crustal thickness?. Is it related to the "A" zone?. This has to be clarified in the captions of Fig. 4b.
- Lines 296-298: Interpretation should go into the discussion
- Lines 309-310: Does "shaded" refer to the areas filled with red?. This has to be clarified.
- Line 369 - 'indicating upper mantle-driven isostatic instabilities'. This is quite a strange phrase because isostasy implies a static, buoyancy equilibrium whereas this paragraph refers to mantle-driving of topography (which implies non-isostatic support)
- Lines 403-406 Can you elaborate on the observed changes in mantle composition?. You mentioned this in the "Tectonic context and motivation" section, but this is an opportunity to link the specific changes to your modelling results.
- Lines 416-420 Shortening this sentence would improve clarity
- Line 469 - advection / diffusion equation is scrambled. It would be helpful to just write these out with equation numbers as it is really difficult to read these equations in-line.

Recommendation

This paper covers a novel and interesting topic and is potentially publishable in Nature Communications. The writing is currently somewhat breathless and that seems to get in the way of readability. Similarly, the figures are elegant but they are cluttered in a way that gets in the way of understanding. We are happy to recommend publication upon addressing comments

Reviewer #3

(Remarks to the Author)

Reviewer #4

(Remarks to the Author)

Review of "Anatomy of a post-subduction collision" by Uluocak et al.

The manuscript by Uluocak and co-authors presents an interesting study on the relationship between mantle thermal structure and various observables from the Arabian–Eurasian collision zone. Overall, the paper is well written and addresses a topic of broad relevance. The potential implications are significant. However, I believe the current modeling approach requires substantial strengthening before the conclusions can be fully supported.

Major comments

1) Modeling strategy

To my understanding, only a single simulation is presented. This limits the robustness of the conclusions. Sensitivity tests exploring variations in the initial thermal structure (inferred from geophysical constraints) are essential. Similarly, the role of mantle rheology, crustal architecture, inherited sutures, and other weak zones should either be explicitly investigated or at least discussed in detail.

2) Model setup and methodology

The modeling approach is not sufficiently detailed in its current form. Model dimensions and resolution should be specified.

Even in instantaneous flow models, runs should be evolved long enough to achieve isostatic equilibration. What timescales were used?

Initial conditions must be clearly introduced: what is the starting thermal and density structure?

A table summarizing all material properties should be provided, ideally in the appendix.

3) Figures and presentation

Figure 2: The contours are unclear. What variable is being contoured? This needs clarification.

Figure 3A: The method used to calculate dynamic topography must be described.

A direct comparison between model outputs and observations is missing. For example, modeled surface velocities should be plotted against GPS data in a single figure. Such comparisons are crucial to convincingly demonstrate the model's validity.

4) Data availability.

If I recall correctly, Nature Communications requires that the data necessary to reproduce the models and generate the figures be made publicly available, for example via Zenodo.

The manuscript presents a substantial body of work and demonstrates the authors' strong understanding of the Arabian–Eurasian collision zone. However, the modeling component requires further work and sensitivity testing to ensure the robustness of the conclusions. Addressing these points would significantly strengthen the study and enhance its impact.

Line per line comments:

L22. I am not quite sure to know/understand what is a “plumelet” at that stage (I am not on top of this literature). As this is likely a key concept of the paper you may want to clarify this for a general audience, right from the start. Is that a small plume? Is it super-adiabatic? Is its buoyancy chemically controlled?

L54. “and the geodynamic model of the upper mantle and lithosphere structures”. Is that based on previous work and reconstructions? You may need a reference here.

§ L51-L65. It is not clear how the model is built from geophysical and geodynamic data at that stage.

L68-L69. “By coupling mantle flow models with thermal 68 and isostatic constraints” this partially answer my previous comment. Maybe this should be moved in previous section.

L73-L74 . “Our results for the first time establish new constraints”, remove “for the first time “, as you have “new constraints” in the same sentence, and also because you use “for the first time” L78.

Fig. 1c. Shouldn't the caption be ΔV_s (or ΔS -wave and not just S-wave) as you display % variation? I am not a geophysicist, but S-wave should be in km/s. If you have % it likely means that this is a variation from a reference model.

L147-148. “The northern branch of the slab subducted beneath Eurasia by creating arc and back-arc volcanism”. Something is wrong in this sentence. I suppose you meant “by creating” -> “created”?

L165. I don't see the Erzurum-Kars plateau “T” on figure 1D.

L192-L195. This should go in method and be expanded (and partly moved to appendix too). How did you integrate the data? Did you invert solely for temperature and you that to define the thermal structure of the model?

What are the densities of your material? How do you choose them? Are they computed from phase equilibrium modelling? How sensitive is your modeling results when propagating error of the inversion? Are you only imposing the initial temperature structure? What are the assumptions in terms of rheologies and material properties for the different crustal and lithospheric units?

Do you define fault, shear zones and weak zones? How are they distributed? Although it does not look like you impose such structures, how important can they be in controlling the stress-strain state of the region? I would suspect that they are rather important.

You also need to provide a table with all used material parameters.

Moreover, I understand you perform instantaneous flow models. Do you account for elasticity? How is isostatic equilibrium achieved? How many timesteps are performed to reach elastic and isostatic equilibrium?

What is your model size and resolution?

L197-L199. At that stage I am still not quite sure to understand what is a plumelet. It seems to me that this is simply a buoyancy contrasts driving upward motion. An upwell?

Fig. 2. What are the iso-contours you are showing on the figure?

Fig. 3a. How do you compute the dynamic topography? This is quite critical to explain how this is computed in your model as this is not trivial especially for non-modelers.

Figs 3b,c Shear stress and Strain rate. Are those invariants? Second invariant the shear stress and strain rates? Otherwise they should be tensors...

Fig. 3d. 3D temperature model with a slice. Are you plotting an iso-surface of the temperature? + a slice? Moreover, there are no yellow dashed lines in Fig. 3d. It needs be to moved in caption 3c.

Fig. 5. Is that realistic to have temperature reaching 2000°C at 300 km depth? Also it seems that large regions of very low temperature are predicted at great depth ~1000°C. Without contouring I have trouble to exactly see the temperature structure, but it seems rather extreme.

L279-L284. Where is the cold anomaly on the figure 3d. I don't see it. Actually, the figure shows really well the trench parallel hot anomaly due to slab break off.

L288-L290. The "Sinking Britis Slab" of figure 5b?

L298. The temperature model should be presented way earlier in my opinion as this is the basis of your modeling! Variation in the temperature model should also be explored to test how sensitive the modelling outcome is.

L301-L305. First, this slab sinking and related upwell and potential erosion are very close to what seems the boundary of the model. How is this constraining the results? Or is it because of how the figure is clipped?

L328-L330. This is rather difficult to see both modelled velocities and GPS velocities are not super-imposed on a figure. Having a figure showing how well the model and the data fit would really help to convince the reader.

L337-L340. This is also difficult to visualize this based on figure 6. Would be better to have a figure super-imposing seismicity and modeled second invariant of the strain rate for instance.

Fig. 6c Given that you are using non-linear T dependent rheology, the discretized viscosity colorbar is an odd choice

L349-L355. How can the southeastern propagation of the deformation be discussed in the light of an instantaneous flow model? I may be misunderstanding here, but from my point of view this can only be explored when performing forward models.

L390-L392. It seems like a bit of a circular reasoning here. As the small scale anomaly are the outcome of using the tomography model as input for the model.

L435- Where is the plumelet coming from? What drives its ascent?

Version 1:

Reviewer comments:

Reviewer #1

(Remarks to the Author)

Dear authors and editors,

I have read carefully the new version of the manuscript and the corresponding answers and it seems the authors have carefully addressed all comments and suggestions given by the reviewers, so in my opinion this version is acceptable for publication in Nat. Comm.

with kind regards

Reviewer #2

(Remarks to the Author)

We are pleased to see the extensive responses of the authors to the comments / recommendations that we provided for this manuscript (and to the other reviews). We are satisfied that the authors have addressed the points we raised in a thoughtful manner and, given the constraints of space and the need to respond to other reviews, we are satisfied with their improvements. We endorse the paper for publication if the editors see fit.

Reviewer #3

(Remarks to the Author)

Reviewer #4

(Remarks to the Author)

Review of "Anatomy of a post-subduction collision" by Uluocak et al.

In the revised version of the manuscript, the authors have addressed all of my comments. The study represents a very interesting contribution, and I thank the authors for their careful and thorough revision.

I have only one minor comment remaining.

Regarding Figures 6b and 6d, I would like to clarify my comment from the previous review, which may not have been sufficiently clear. When I wrote, "First, this slab sinking and related upwelling and potential erosion are very close to what seems to be the boundary of the model," I was questioning the extent to which the lateral boundary conditions may influence the model behavior near the margins.

For instance, in Figure 6e, the flow in the top left and top right corners appears to rotate rapidly and become parallel to the left and right model boundaries. My question is whether this behavior reflects a boundary effect. If so, how does this influence or constrain the interpretation of the modeled flow patterns and associated processes near the boundaries?

Nicolas Riel

Response Letter to Editor

We sincerely thank you and Reviewers for their insightful reviews of the manuscript. We agree that these additional clarifications are helpful in improving the explanation of our approach. A number of changes have been done in the revised manuscript. In particular, we have made the following primary modifications:

- 1) The Introduction has been revised to include a new Figure 2 that more clearly illustrates the tectonic evolution of the region as suggested by Reviewers 1 and 2. Corresponding references for this conceptual model have been added in the revised manuscript.
- 2) We have modified figures (revised Fig 1d, Fig 3, Fig. 7) to show variations, symbols and results more clearly as suggested by the Reviewers. Related captions have been corrected/modified to explain figures and our findings clearly.
- 3) In response to Reviewer comments, we have added new references to the manuscript. To accommodate these within the reference limit, we removed one reference on local-scale changes (Oyan et al., 2023) and some modeling references (Uluocak et al., 2016; 2019), whose context is covered by other cited works.
- 4) As suggested by Reviewer 4, the Method section has been expanded with additional information, references and Table to clarify material parameters, modeling time and space resolutions. We think that the revised manuscript will make this modeling strategy more explicit for a broader audience.
- 5) We clarified Data Availability section in the response letter. The full suite of input parameters is detailed in the following link (<https://doi.org/10.5281/zenodo.12568287>) that has already been shared publicly by Uluocak, et al. (2024, e2024GC011639) and Uluocak (2024, <https://doi.org/10.5281/zenodo.12568287>). All codes used in the numerical calculations are also available in open-sourced ASPECT directory (<https://aspect.geodynamics.org/>). Seismic data used in modeling is also available in Piromallo & Morelli (2003), as stated in the text (e.g., Introduction, Method). Further, we state that ESU, the corresponding author, will gladly assist anyone who needs help compiling these open sources. The core objective of this study is to utilize a proven/confirmed numerical modeling methodology to investigate complex processes in collisional settings by considering standard space and reference limitations of the journal. We think this strategy can allow us to focus on elucidating deformation mechanisms and to effectively communicate these new findings to not only modelers but also a broad readership of Nature Communications.
- 6) We thank Reviewers for also bringing our attention to minor corrections, we have addressed these and typos in the text to ensure they are fixed in the revised manuscript as shown in the tracked-changes file. We believe the revised manuscript presents a much clearer and more cogent contribution representing our original new research.

Sincerely yours.

On behalf of the Authors

Dr. Ebru Şengül Uluocak (sengul@gfz.de, ebusengul@comu.edu.tr, ebusengul@gmail.com)
(Corresponding Author)

REVIEWER COMMENTS AND RESPONSES

Reviewer #1 (Remarks to the Author):

-I read this manuscript with great interest as it presents a novel perspective on the Arabia-Eurasia collision zone, drawing upon a synthesis of geological and geophysical data, as well as thermo-mechanical modelling. This new perspective is supported by the data, and I recommend that it be considered for publication in Nature Communications. However, I believe that some points still need to be clarified before it can be accepted.

Response: We thank the reviewer for their careful review and comments. We have modified the manuscript with additional explanations, figures, and references. Our point-by-point responses are listed below. Also please see the Revised tracked file that highlights tracked changes and revised file for numbers of the lines.

-Firstly, it seems that the authors are pitting their model against a 'traditional model', but the latter is not clearly explained in the present paper. In my view, there are several theories regarding the evolution of the Caucasus region. Some previous papers suggest a significant role for delamination (cf. Grosjean et al., 2022), a process which is not clearly discussed or taken into account in this paper. Some papers have already proposed the contribution of the deep asthenosphere to the magmatism of the area (e.g. Soltanmohammadi et al., 2018), thus excluding any purely 'collisional' component.

Previous models did not strictly fall into a single category, and I agree that the horizontal mantle flow or 'plumelet' model proposed here is novel, as previous models assumed that the mantle flowed mainly vertically.

Response: Thanks to the reviewer for raising this point. We agree that the Arabian-Eurasian collisional zone has been the subject of numerous studies with diverse datasets aiming to characterize processes both prior to and during its post-subduction phases. Given the journal's word and reference limits, a significant challenge for our manuscript was to concisely synthesize this large body of previous work. For this reason, in the first section, we focus primarily on neotectonics and present-day observations with brief information on the tectonic evolution of the region (with appropriate references), while we discuss previous models in the last section by also citing Grosjean et al. (2022).

Contrary to models that attribute deformation in the region solely to lateral mantle flow, boundary forces, or small-scale vertical convective flows linking to the slab dynamics (e.g., delamination/detachment/break-off; Keskin, 2007; Göğüş, & Pysklywec, 2008; Bewick et al., 2022), we propose a more complex framework for collisional settings. Namely, we posit that deformation mechanisms involve the spatial and temporal interplay of both vertical mantle impulses and lateral forces owing to plumelet propagation. Yet, we agree with the reviewer that it is an oversimplification to categorize all prior works under a single, unifying deformation hypothesis. It was not our intention to present them as such, and we thank the reviewer for giving us the opportunity to clarify this point in our revised manuscript. Therefore, we have improved the text with additional explanations, references and a new Figure 2 (please see the following comment) to properly acknowledge diversity of existing models. We believe that Figure 2 and the revised manuscript (we thank for your contributions) will help readers to better understand the tectonic evolution of the region, including the Caucasus segment of the collision. In the Introduction, we've cited Faccenna et al., (2013), Rolland (2017), Rolland et al (2012), Grosjean et al. (2022) (but not Avagyan et al., 2010 suggested by Rev.1 because of space limitations) to develop our arguments-- the important role of mantle dynamics in plate consumption, magmatism and partial melting in the region. These revisions can be followed below and in the tracked changes file (both in Introduction and Discussion):

Introduction, Tectonic context and motivation: “The lithospheric heterogeneity documented beneath the region (Uluocak et al., 2021) can be attributed to Tethyan plate orogeny, as evidenced by numerous observations. The northern branch of the slab subducted beneath Eurasia by creating arc and back-arc volcanism, mostly dating from the Jurassic to Eocene (e.g., Şengör & Yılmaz, 1981; Adamia et al., 2011; Rolland et al. 2012; Rolland, 2017 and references therein). This part of the slab broke off around the Early Eocene, triggering crustal shortening in the Greater and the Lesser Caucasus by the Oligocene (Avdeev & Niemi, 2011; Cowgill et al., 2016). Concurrently, the southern branch (Bitlis-Zagros slab) initiated northward subduction, and the collisional front started to form in response to the Eocene-Oligocene Zagros slab roll-back with the mantle window (slab gap) opening between the Zagros and the Bitlis slabs (Rabiee et al., 2020 and references therein). Subduction jumped northward (~ Eocene, e.g., Rolland et al., 2012), and the closure of the oceanic basin persisted due to the ongoing collision between the Anatolian-Southern Armenian Block and Pontides active margin (Rolland et al., 2012; Shiva et al., 2025). This phase was alternatively characterized by the initiation of south-dipping subduction of the Caucasus Basin along the arc (e.g., Rolland 2017). Terminal closure along the Bitlis-Zagros suture (Late Oligocene–Middle Miocene, Fig. 2a) may have led to slab detachment beneath eastern Anatolia at ~25-10 Ma (Keskin, 2007; Şengör et al., 2008; Rabayrol et al., 2019 and references therein, Fig. 2b). Ongoing convergence reactivated Pliocene uplift, folding in the Kura Basin with crustal shortening in the southern Greater Caucasus, intensified exhumation across the Greater Caucasus, and strike slip deformation to the south (e.g., North and East Anatolian faults, Figs. 1a, 1b, & 2c) (Adamia et al. 2011; 2017; Avdeev & Niemi, 2011; Cowgill et al. 2016; Rolland, 2017; Ismail-Zadeh et al., 2020; Molin et al., 2023). Consistent with numerous studies, such as river network analyses (Molin et al., 2023), the initial break-off of the southern Neotethys slab beneath the Bitlis-Zagros suture zone could have triggered mantle upwelling, propagating from SW to NE beneath the Arabian Plate --a plumelet (Uluocak et al., 2024, Fig. 2d). Among various slab deformation models, removal mechanisms involving spatiotemporal overlapping processes (Uluocak et al., 2024) can be primarily characterized by: i) wholesale or progressive Bitlis-Zagros slab delamination along the Arabian collisional front in the south (Keskin 2007; Memiş et al., 2020 and references therein); and ii) delamination beneath the Transcaucasus (e.g., Shiva et al., 2025 and references therein) accompanied by northward plate tearing toward the Greater Caucasus (e.g., Mumladze et al., 2015). Although 3D heterogeneities beneath the region are still debated, magmatic records provide evidence of mantle-crust interactions as a consequence of a discontinuous/semi-episodic closure of the Neotethys Ocean (e.g., Rolland et al., 2012).”

-The Tectonic context and motivation section is well-written, but is hard to follow due to the absence of a figure that provides context for the tectonic evolution of the study area.

Response: The revised text has been enhanced by the inclusion of an additional figure - Figure 2- showing the generalized tectonic evolution of the region (as also recommended by Rev.2).

Redacted

-On the other hand, in the 'Results' section, the authors immediately propose a model without first presenting the results in a cautious or analytical manner. For example, at lines 197–199, they claim that 'Mantle dynamics are dominated by a focused, rapid, SW–NE-directed inflow from the Arabian foreland to the eastern Greater Caucasus—a plumelet—penetrating through the complex morphology of the post-collision environment'. While I believe this interpretation

is entirely plausible, it is crucial to present results more factually and demonstrate how they support the plumelet model.

Response: Upper mantle flows propagate from SW to NE in the Arabian-Eurasian collision zone, a region encompassing much of the western part of our study area. A similar flow was introduced as a plumelet in previous work (Uluocak et al., 2024) that synthesizes/reviews deformation models and observations both supporting and contesting the plumelet hypothesis (e.g., Ershov & Nikishin, 2004; Doi: 10.4236/ijg.2014.55047). While we previously cited this study in our tectonic context with Figure 1c (and new figure; Figure 2d), we have now expanded our discussion in the revised Introduction and Discussion sections to further explain plumelet dynamics. Namely, although the plumelet is not a new concept (as stated in the text clearly), we claim that impacts of a plumelet in collisional settings, such as plumelet-induced plate erosion, low viscosity zone and its seismogenic implications are novel. We have modified revised text to make the narrative clearer:

Introduction: “By coupling mantle flow models with thermal and isostatic constraints, we reveal not only a SW-NE oriented plumelet from the Arabian foreland injecting into the subduction system (Uluocak et al., 2024), but also 3D thermomechanical erosion of the upper plates and the active Neotethys slab deformation along the southern collisional front.

...Consistent with numerous studies, such as river network analyses (Molin et al., 2023), the initial break-off of the southern Neotethys slab beneath the Bitlis-Zagros suture zone could have triggered mantle upwelling, propagating from SW to NE beneath the Arabian Plate--a plumelet (Uluocak et al., 2024).”

Further points should be clarified:

-What role do vertical structures on the lithosphere scale play in propagating the plume? The inception of horizontal flow within a structure featuring vertical slabs requires zones of weakness inherited from the subduction-early collision history (step faults etc).

Response: We aim to define the complex morphology of the plumelet's propagation channel and its interaction with deep basins, high plateaus, and uppermost mantle cold anomalies (e.g., the Pontide-cold). We show thermomechanical erosion caused by the plumelet (e.g., the actively sinking cold material beneath the Arabian collisional front), by linking these processes to the region's subduction and early collision history. We further identify deep-seated inherited plate boundaries (e.g., the Aras Fault) as the inter-plate deformation between converging blocks. Because of limitations of the journal, further discussion will have to be reserved for subsequent work. Thank you for your comment and contributions.

- The so-called 'Bitlis slab' may also be the result of delamination, because the Bitlis subduction zone is characterised by HP metamorphism at 74–71 Ma (e.g. Rolland et al., 2012), which likely resulted in sinking at very deep levels of the mantle (deeper than 300 km) and should have been N-dipping (no volcanic arc developed towards the south). This tectonic zone is generally considered to be complex and composed of numerous tectonic blocks and sutures. Therefore, the 'closure of Neotethys' should not be considered a progressive process, but rather a discontinuous 'step-by-step' process in which basins were successively closed. This explains why the timing of 'collision' is disputed in this region.

-Rev. 1: maybe this has nothing to see with a Bitlis slab (more likely N-dipping...), rather delamination?

Response: We thank the Reviewer for these comments on the Bitlis feature. We agree that uppermost mantle structures beneath the region are quite ambiguous and need to be clarified

through high resolution observations and further evolutionary numerical models. In general, these anomalies (both Pontide-cold and Bitlis slab anomalies as termed in the manuscript) are interpreted as “Bitlis slab fragments” in observations and previous deformation models (e.g., Keskin 2007; Portner et al., 2018; Memis et al., 2020). In other words, relatively small-scale/shallow (not deeper than ~300 km)/cold bodies beneath the collision zone, extending from the northern Arabian Platform to the eastern Pontides, are interpreted as a remnant part of fully detached Bitlis slab fragments. However, to open up the interpretation in the revised manuscript we now describe the southern anomaly as “detached slab or delaminated Bitlis fragments”. We also interpret two groups of anomalies spatially separated (beneath the arc and the Arabian Platform) within the complex Cenozoic tectonic concepts of the region: i) arc-along deformation pattern (drip-like deformation) in the north (beneath the Pontides-Transcaucasus arc); ii) fully and partly detached (active sinking) Bitlis slab beneath the thick Arabian lithospheric mantle connected with the eastward propagation of the plate tearing. This new concept may create a link between active eastern Zagros subduction zone along the Arabian collisional front/Bitlis-Zagros suture zone. Observations, including magmatism and seismic tomography datasets, indicate a gap between Zagros subduction and its eastern part along the Bitlis-Zagros suture zone -i.e., Bitlis Slab- in response to postulated Eocene-Oligocene Zagros slab roll-back with a mantle window (slab gap) opening between the Zagros and the Bitlis slabs (e.g., Rabiee et al., 2020), which we believe supports our interpretation here. Following the Reviewer's advice, we revised the text to clearly express tectonic evolution and our findings:

Results: “This anomaly differs from the NW Zagros depression area (P2; Figs. 6d-e) in that it is decoupled from the upper plate, implying fully detached cold material- termed as a part of a detached slab or delaminated Bitlis fragments - sinking beneath the North Arabian Platform.”

Tectonic context and motivation: “Magmatic records provide evidence of mantle-crust interactions as a consequence of a discontinuous/semi-episodic closure of the Neotethys Ocean (e.g., Rolland et al., 2012).”

Discussion: “These results reconcile previously disparate observations - from shallow (< 300 km), fast seismic velocities (at ~250 km depth; Koulakov et al., 2012) to overlying slow seismic anomalies (e.g., Koulakov et al., 2012; Civiero et al., 2023; the plumelet Uluocak et al., 2024)- into a unified geodynamic framework validated by different numerical models (Uluocak et al., 2021) based on recent seismic tomography data (Kounoudis et al., 2020).”

-Overall, the text is well written, but some sections could be improved (I have underlined some examples in the text). Please avoid self-satisfaction (e.g. 'Our comprehensive geological, geophysical and geodynamic analyses illuminate the tectonic anatomy...').

Response: Thank you for your detailed review. We have edited the text again, trying to correct such instances—including the underlined examples and the quoted passage (please see the tracked changes file).

-The figures are well drafted and informative. However, when dealing with seismicity (Fig. 6), it is difficult to understand the clustering of seismicity. It might be better to present a representative vertical section showing the seismicity in relation to the crustal levels.

Response: We agree with the reviewer regarding the complexity of earthquake distributions, although we already show seismicity more clearly in Figure 1b. In former Figure 6c, we only aimed to focus on subcrustal seismic events in a specific part of the study area. Following the

reviewer's suggestion, in the revised manuscript we have added a new figure -Figure 7d- that specifically shows distribution of both crustal and subcrustal earthquakes (Magnitude of ≥ 2.5 , USGS, 1900-2025) as a longitudinal profile for the zone between 34°N-35°N (south of profile-P3). Revised Figure 7 (former Fig. 6) has also been improved by an inset to clarify directions (east, west, north, & south, depth) used in revised Fig. 7b-c. We think that these additional figures, as well as Figure 1b, clearly image how the seismically active region in the east differs from the west of the collisional front, which is subject to thermomechanical erosion.

-I also think the title does not fully reflect the content, as it should emphasise the 'plumelet' hypothesis.

Response: We thank the reviewer and carefully considered their suggestion, but respectfully we choose to retain the original title. We wish to emphasize that our study focuses on the anatomy of post-subduction collisional settings with complex deformation patterns.

-Not clear: "...the plumelet may reworks the subducted lithosphere actively converging with Eurasia"

-the plumelet supposedly causes break-off? "is that really considered "collision related deformation"?"

-is there just one traditional model (then what is it?) or a multiple pannel of proposed models?
"In the overall tectonic anatomy of the complex plate boundary, during its migration from deep mantle source(s) in the south, the plumelet reworks the subducted lithosphere actively converging with Eurasia. As well as causing collision-related deformation (e.g., slab breaking-off), the plumelet induces thermomechanical erosion of the foundering slab at the active collisional margin. The results discover a more intricate multi-scale behavior to the post-collisional tectonic boundary than the traditional model..."

Response: The multiple previous deformation models (including numerical experiments) mostly focus on the Neotethyan slab dynamics involving advance, retreat, and eventual break-off of the slab to explain complex post-subduction tectonics of the region. Here, we aim to clarify forces caused by mantle dynamics in the form of relatively large- (plumelet) and small-scale convective cells (plumelet-induced upwellings) and provide a link between tectonic evolution and present-day deformation patterns. In addition to the thermal and isostatic constraints shown in the text, our results are supported by numerous observations, such as intense magmatism with mantle-sourced fingerprints and high plateau evolution (pls see Fig. 2 in Uluocak et al., 2024, below). All indicate that convective forces may have been active since the Pliocene and could have formed post-subduction tectonics in the Arabian-Eurasian collisional zone. We have rephrased the passage quoted by the Reviewer to clarify:

Discussion: "In the tectonic anatomy of the complex plate boundary, during its migration from deep mantle source(s) in the south, the plumelet may rework the subducted lithosphere actively converging with Eurasia. That is, the plumelet plays an active role in plate consumption, as demonstrated by thermomechanical erosion in the overlying plate and sinking slab along the active collision margin and by drip-type removal beneath the volcanic arc. The results reveal a more intricate multi-scale behavior of the post-collisional tectonic boundary than the various traditional models (e.g., Keskin, 2007; Memiş et al., 2020) based solely on slab dynamics and boundary forces."

-what is a foundering slab?

Response: We used this term referring to slab detachment processes in a broad sense, but the phrase has been deleted to avoid possibly being misleading.

-Plate tectonic collision zones can experience prolonged crustal and upper mantle

deformation that defines *often* (marked by Rev) post-subduction tectonics in continental orogenic settings.

Response: Removed.

-I hope these comments are useful. With best regards.

Response: We sincerely thank you for your careful review and constructive comments.

-refs

Grosjean, M., Moritz, R., Rezeau, H., Hovakimyan, S., Ulianov, A., Chiaradia, M., & Melkonyan, R. (2022). Arabia-Eurasia convergence and collision control on Cenozoic juvenile K-rich magmatism in the South Armenian block, Lesser Caucasus. *Earth-Science Reviews*, 226, 103949.

Soltanmohammadi, A., Grégoire, M., Rabinowicz, M., Gerbault, M., Ceuleneer, G., Rahgoshay, M., ... & Benoit, M. (2018). Transport of volatile-rich melt from the mantle transition zone via compaction pockets: implications for mantle metasomatism and the origin of alkaline lavas in the Turkish–Iranian plateau. *Journal of Petrology*, 59(12), 2273-2310.

Rolland, Y., Perincek, D., Kaymakci, N., Sosson, M., Barrier, E., & Avagyan, A. (2012). Evidence for ~ 80–75 Ma subduction jump during Anatolide–Tauride–Armenian block accretion and ~ 48 Ma Arabia–Eurasia collision in lesser caucasus–East Anatolia. *Journal of Geodynamics*, 56, 76-85.

Alizadeh Noudeh, S. Yann, R., Magali, R., Delphine, B., Philippe, M., Arthur, I., ... & Rahgoshay, M. (2025). Geochronological, isotopic and petrogenetic investigations of Cenozoic Volcanic rocks in the Talysh Massif, NW Iran: Insights for the Eocene magmatic flare-up. *Lithos*, 496, 107954.

Response: The revised text has been modified by adding the suggested references, we thank the Reviewer for their contribution.

Reviewer #2 (Remarks to the Author):

Anatomy of a Post-Subduction Collision

****Authors:**** Ebru Şengül Uluocak, Russell N. Pysklywec, Claudio Faccenna & Taylor Schildgen

This paper addresses the problem of understanding mantle dynamics in continental collision zones where the driving oceanic slab has detached or broken off from the overriding plate. These post-collisional systems are characterized by prolonged crustal and upper mantle deformation that can persist long after the initial convergence and slab consumption has ceased. The Arabian-Eurasian collision zone exemplifies such complexity, where the Neotethyan oceanic slabs have undergone various stages of subduction, break-off, and foundering, leaving behind a fragmented lithospheric architecture. Understanding what drives continued deformation in these post-subduction environments remains a fundamental challenge in tectonics, as traditional plate-driving forces are absent yet significant surface deformation, volcanism, and seismicity persist.

Overview

The manuscript is well-written and organised and it is of significance because it showcases how mantle dynamics strongly influences strain rate, stress, topography, and mantle flow of post-collisional settings. It also demonstrates how tectonic activity post-dates the interruption of subduction due to the effect of mantle dynamics and provides a framework for the integration of geodynamic modelling with geophysical observables.

The approach is based on correlating various geophysical observables - seismic velocities, topography, crustal thickness, and seismicity patterns - with their 3D thermomechanical model results. The authors systematically demonstrate how regions of high V_p/V_s ratios correspond to magmatic arcs and elevated topography, how cold mantle anomalies correlate with thick lithospheric roots and deep basins, and how modeled strain rates align with observed seismicity distributions. It is these (positive) correlations that form the empirical foundation for validating their model predictions and serve as the primary evidence supporting their interpretations.

Response: We sincerely thank Reviewer 2 for their careful review.

-Another central concept is the "plumelet" which is defined by the authors as the migration of upper-mantle plume material into the region. **It may be drawn in by flow in the region or by its own buoyancy.** The plumelet is proposed as a flow of hot mantle material from the Arabian region to the eastern Greater Caucasus. They propose that, from their 3D thermomechanical modeling, this lateral mantle flow **drives** regional deformation, influences slab break-off processes, **causes** thermomechanical erosion of overlying plates, and creates the observed dynamic topography patterns across the Arabian-Eurasian collision zone. There are a number of other areas where subduction influences mantle flow - for example the paper by Miller & Becker (<https://doi.org/10.1130/G34959.1>) argues that uplift of the Atlas is enhanced by flow of plume material around the foundering Alboran slab. Or indeed the paper by Lanari et al (2023) which has some author overlap with this paper. Are the other notable examples that would help to establish this plumelet hypothesis and generalise it ?

Response: We appreciate this constructive feedback. Unfortunately, due to the journal's reference and space limitations, we were unable to cite many other articles in the manuscript. However, we have rephrased this section with suggested reference (Miller & Becker) by removing Lanari et al., 2023 (due to the journal's reference limitation)- that may help build and generalize the plumelet argument.

-A fundamental question raised by this interpretation is whether the plumelets ****are**** actively driving the observed tectonic signals, or whether they represent more passive structures that are being drawn into the region by the process of slab breaking off and foundering. The distinction between active versus passive roles has significant implications for understanding the causal relationships in post-collisional tectonics and the broader applicability of the plumelet concept to other collision zones. Is the plumelet genuinely causing these effects? Can it be determined convincingly through the models?

Response: Based on our analyses, we believe that mantle dynamics play an active role in driving tectonics in the region. As well as the thermal and isostatic constraints provided in the manuscript, it can be concluded from previous studies (e.g., Fig. 2 by Uluocak et al., 2024 and references) that convective forces may have been active at least since the Pliocene and formed post-subduction tectonics in the Arabian-Eurasian collision zone. The link between present-day observations and post-subduction related deformation patterns imaged in our work here provides support that mantle flow is playing an active role in the tectonics of the region. This approach gives a new and integrated assessment for the collisional settings. We revised the text to make clearer our findings and conclusion as follows (Discussion):

Redacted

Discussion: “In the tectonic anatomy of the complex plate boundary, during its migration from deep mantle source(s) in the south, the plumelet reworks the subducted lithosphere actively converging with Eurasia. That is, the plumelet plays an active role in plate consumption, as demonstrated by thermomechanical erosion in the overlying plate and the sinking slab along the active collision margin, and by drip-type removal beneath the volcanic arc. The results reveal a more intricate multi-scale behavior of the post-collisional tectonic boundary than the traditional models (e.g., Keskin, 2007; Memiş et al., 2020) based solely on slab dynamics and boundary forces.”

-In Figure 1, the main magmatic units are shown; however, it would be beneficial to provide information about the composition of these volcanic units. Therefore, a figure showing the tectonic evolution, the location of magmatic units and their chemistry is needed. This is very important for readers who are not familiar with the geology of the study area.

Response: The revised manuscript has been improved with an additional figure (new Figure 2) and references to clarify the tectonic evolution of the region. Because of the journal's limitations, providing another figure with discussion on magmatism (chemistry, ages and even samples' locations) seems not feasible/possible. Yet, following the reviewer's suggestions, we have improved the text with additional references and explanation:

Tectonic context and motivation: “Post-collisional volcanism initiated (around in the early–mid-Miocene) on the Erzurum-Kars Plateau (locations shown by “T” in Fig. 1d) and migrated south and eastward, towards the Bitlis-Zagros Suture and the Armenian border, respectively, consistent with radiometric ages (e.g., Lin et al., 2020; age compilation in Memiş et al., 2020). The late Miocene-Quaternary calc-alkaline to alkaline magmatism is documented across the collisional front with ages decreasing westward and subduction signatures weakening southward, towards the Bitlis-Zagros suture (Keskin, 2007 and references therein). Based on chemical and petrological analyses of widespread alkaline lavas active since the Late Cenozoic in the Turkish-Iranian Plateau, it has been argued that the volcanism may be linked to a low seismic velocity zone/body extending to depths of ~300 km (Soltanmohammadi et al., 2018).”

-In general, figures are very pretty but not always easy to follow. Figure 5 is quite straightforward, but Figure 6a is not. Figure 6b is difficult to see compared to figure 6c. We know the 3D renderings are a lot of work to produce but in static form they are less useful than the cross sections (sorry).

Response: We understand the complexity of the figures raised by the Reviewer. For the revised manuscript we have improved figures (Figures 1d, 3, 5, 6, 7) with different perspectives/map that may help to clarify our results here. Namely, we have improved revised Figure 3 (previous Fig. 2) with additional figure (Figure 3a) to clarify locations of 3D anomalies and profile in revised Figure 3b and revised Figure 7. We also improved Figure 7 (former Fig.6) with an additional figure and an inset. Figure 7d images crustal and subcrustal earthquake distributions along the profile P3 for a zone between 34°N-35°N (“The longitudinal distribution of earthquakes between 34°N-35°N (USGS, 1900-2025). Red stars indicate thermomechanically eroded zones.”). An inset figure shows cartesian coordinates to make the 3D orientations clearer.

-Figure 6 is helpful because it integrates all the observables and the results from the numerical modelling. However, the mantle structure of the study area is quite complex and the 3D geometry of the plumelet propagation channel is not fully clear. We suggest you could draw your interpretation of this geometry in Figure 6b, and this will help the reader to better visualize how it correlates with variations in stress, strain rate, dynamic topography and V_p/V_s ratios.

- Lines 333-336: The vectors in Figure 6 can be improved. This would facilitate the comparison with the observed GPS velocity field. Also, Figure 6 shows a zoomed region, but the extent of this zoomed region is not clear. It would be useful to include an inset with a regional map showing the location of this region.

Response: We agree with the reviewer that cross-sections projecting results onto a 2D plane are easier to follow. However, they can also be oversimplified and potentially misleading. Thus, we have included both 2D and 3D images to accurately represent the complex deformation in the collisional zone. In fact, we aim to make this complexity visible in terms of deformational patterns in the collisional setting.

In the text, as well as 3D views (revised Figure 3, 4 and 7) in the manuscript, we show plumelet propagation and geographic locations in the revised Figure 3 that includes cross-section of P1 also shown in revised Figures 6 and 7. Namely, in response to Reviewer's concern regarding the link between geographic locations and anomalies, the revised Figure 3 has been improved with a new map (revised Figure 3a) showing locations of the profile (P1) and various features (names, arc, plateaus). By following the suggestions, we have also improved revised Figure 7 (previous Fig. 6) by adding new figure (revised Fig. 7d) to establish the link between our results, the locations of deformations, and seismicity. Furthermore, in order to avoid additional complexity of Figure 7, an inset has been added in revised Figure 7 to clarify the coordinate system and orientations of 3D images in Figures 7a-c. Captions and text have been revised accordingly. Due to the journal's space limitations, we address Figure 1 to compare GPS orientation with our results. We thank the reviewer for this comment, which has helped us improve the manuscript.

Detailed Feedback

- Pn and Sn are quite well known but not universally, it would be useful to add a short explanation

Response: This part has been improved as follows:

Tectonic context and motivation: “Comparable high-velocity anomalies at shallow depths ($\leq \sim 150$ km) are observed beneath the Black Sea, Mesopotamian-Zagros foreland basins, and Zagros fold-thrust belt, consistent with high Pn seismic wave velocity perturbations in these regions ...

In contrast to high amplitudes of guided seismic waves (Pn and Sn) that travel in the lithospheric mantle, crustal seismicity and slow seismic velocities correlate well with zones of high Sn and Pn attenuation, indicating thinned or absent lithosphere (≤ 50 – 90 km) beneath the East Anatolian, TGA, the North Iranian plateaus and ...”

- Lines 86-89: In the captions of Figure 1d, are the red triangles showing the magmatic units or active volcanoes?

Response: Volcanoes. The caption has been revised to clarify.

- Lines 149-150: It is worth mentioning which observations this refer to (geological, geophysical, geodynamic) ?

“The northern 148 branch of the slab subducted beneath Eurasia by creating arc and back-arc volcanism, mostly 149 dating from the Early Cretaceous to Eocene (e.g., Şengör & Yılmaz, 1981; Adamia et al., 2011).”

Response: The tectonic context and motivation section has been revised with an additional figure (Fig.2), explanations, and related references to clarify evolution history and observations. This revised description addresses geochronological, isotopic and petrogenetic investigations of volcanics of the region (in an abbreviated form owing to the journal’s space limitations).

Tectonic context and motivation: “The northern branch of the slab subducted beneath Eurasia by creating arc and back-arc volcanism, mostly dating from the Jurassic to Eocene (e.g., Şengör & Yılmaz, 1981; Adamia et al., 2011; Rolland et al. 2012; Rolland, 2017 and references therein)”

- Lines 204-205: Figure 2 would benefit from adding a label with the location of the Scythian plate.

- Line 228: Is there any reason for Figure 2 to display the north at an orientation different from all other figures?. Although you include the labels of the main tectonic domains, it is confusing.

Response: In the revised manuscript, Figure 3 (previous Fig. 2) has been modified for clarity. Particularly in this figure, we aim to show plumelet propagation with convection vectors clearly with a view that is different from the revised Figures 6 & 7. We added a new topographic map (revised Figure 3a) showing the profile location (P1) with relevant place names. The location of the Scythian Plate is now included in the revised Figs. 3a and b. Also, the figure captions have been revised in accordance with the Reviewer's comments.

- Lines 221-222: differences of less than 1MPa do not seem that “high”

Response: This statement has been modified in the revised text:

Results: “The flow modelling indicates regional deformation zones in the central and eastern Greater Caucasus extending to the stable Scythian Plate, marked by visible lateral contrast of low shear stresses in the crust (≤ 1 MPa; Fig. 4b)...”

- Lines 249-251: you mentioned “crustal velocity ratios”; however, you didn’t mention the crustal depth or depths considered in the contours.

Response: To avoid complexity in revised Figure 5 (previous Fig. 4), only average values of the V_p/V_s ratios were obtained at depths of 20 km and 40 km (Zhu, 2018). These depths have been added to the revised caption for Figure 5.

Figure 5. “... Average of V_p/V_s ratios for the crust (at 20 km and 40 km depths) and V_p/V_s ratios for the depth of 100 km (Zhu, 2018).”

- Lines 274-275: What is the purpose of the vertical line at ~ 35 km of crustal thickness?. Is it related to the “A” zone?. This has to be clarified in the captions of Fig. 4b.

Response: That is right, it is related to south Armenian highland; the caption of figure has been corrected to clarify this.

Figure 5b: “A” stands for the Armenian volcanic highland with relatively thin crustal thickness (gray dashed line) and is interpreted as concerning partially melted lower crustal features (Lin et al., 2020).”

- Lines 253-254: “corroborates” is not the most suitable word for your explanation

Response: Thank you for the correction, it has been rephrased:

Results: “This result demonstrates a direct mantle-crust linkage in concordance with crustal and subcrustal thermal anomalies beneath young volcanic centers (Fig. 5a) and our 3D temperature model (Figs. 3b & 4d).”

- Lines 283-284: What does “Pontide cold” mean?

Response: We use this term to indicate relatively shallow localized cold bodies by following the same terminology in Uluocak et al. (2019).

- Lines 296-298: Interpretation should go into the discussion

“ We interpret an active slab breaking off beneath the East Greater Caucasus with a cold thick root of stable Scythian Plate at the northern boundary of the collision (P2 in Fig. 5e).”

Response: The statement has been rephrased as follows:

Results: “We define an active slab breaking off beneath the East Greater Caucasus with a cold thick root of stable Scythian Plate at the northern boundary of the collision (P2 in Fig. 6e).”

- Lines 309-310: Does “shaded” refer to the areas filled with red?. This has to be clarified.

Response: Thank you for the correction, it has been done:

Figure 6. “... observed topography (gray) along profile..”

- Line 369 - 'indicating upper mantle-driven isostatic instabilities'. This is quite a strange phrase because isostasy implies a static, buoyancy equilibrium whereas this paragraph refers to mantle-driving of topography (which implies non-isostatic support)

“Our integrated analysis (Fig. 4) demonstrates a systematic correlation between high average V_p/V_s ratios (≥ 1.8) at elevated regions (≥ 1 km) and relatively thin crusts (≤ 40 km), indicating upper mantle-driven isostatic instabilities. The Turkish-Georgian-Armenian (TGA) volcanic provinces, East Anatolian Plateau, western Greater Caucasus, Talesh (Azerbaijan), and Urumieh-Dokhtar magmatic zone are defined by isostatically under-compensated topography with thermally heated crustal and uppermost mantle structures. “

Response: Agreed--these variations show non-isostatic support from mantle; to clarify the statement ‘isostatic instabilities’ has been rephrased as ‘non-isostatic support’ in the revised manuscript.

Discussion: “Our integrated analysis (Fig. 5) demonstrates a systematic correlation between high average V_p/V_s ratios (≥ 1.8) at elevated regions (≥ 1 km) and relatively thin crusts (≤ 40 km), indicating upper mantle-driven non-isostatic support.”

- Lines 403-406 Can you elaborate on the observed changes in mantle composition?. You mentioned this in the “Tectonic context and motivation” section, but this is an opportunity to link the specific changes to your modelling results.

“This interpretation is409 corroborated by relatively local crustal thinning, thermal heating, and magmatism with mixed410 geochemical signatures documented in the Armenian volcanic highland (Zabelina et al., 2016;411 Neil et al., 2015; Bewick et al., 2022). Although region-specific geodynamic models (e.g., Curie412 et al., 2015) provide additional kinematic constraints, our drip hypothesis establishes a useful413 approach for interpreting both plumelet and Pontide-cold anomaly and possible related414 magmatic arc cycle with observational (geological-magmatic-tectonic) signatures changing415 from arc to an intraplate deformation above former subduction zones within continental416 collision systems (e.g., Burg, 2011; Allen et al., 2013; Ducea et al., 2015; Maierová et al., 2025).”

Response: We agree with Reviewer 2, but since we have exceeded the 5000-word limit and have already made the relevant explanation in the Introduction, expanding on this part is just not feasible.

- Lines 416-420 Shortening this sentence would improve clarity

Lines 416-420 in text with shifted lines: “Although region-specific geodynamic models (e.g., Curie et al., 2015) provide additional kinematic constraints, our drip hypothesis establishes a useful approach for interpreting both plumelet and Pontide-cold anomaly and possible related magmatic arc cycle with observational (geological-magmatic-tectonic) signatures changing from arc to an intraplate deformation above former subduction zones within continental collision systems”

Response: As suggested by the Reviewer, we have shortened this sentence to make the passage more succinct:

Discussion: “Although region-specific geodynamic models (e.g., Curie et al., 2015) provide additional kinematic constraints, our drip hypothesis establishes a useful approach for interpreting both a plumelet and Pontide-cold anomaly. We therefore claim that there is a Pontide-cold-related magmatic arc cycle here supported by observational (geological-magmatic-tectonic) signatures changing from arc to an intraplate deformation above former subduction zones within continental collision systems....”

- Line 469 - advection / diffusion equation is scrambled. It would be helpful to just write these out with equation numbers as it is really difficult to read these equations in-line.

Response: This section has been revised with the expressions on separate lines (with equation numbers) to make the governing equations clearer.

Recommendation

This paper covers a novel and interesting topic and is potentially publishable in Nature Communications. The writing is currently somewhat breathless and that seems to get in the way of readability. Similarly, the figures are elegant but they are cluttered in a way that gets in the way of understanding. We are happy to recommend publication upon addressing comments

Response: We appreciate Reviewer 2 for the providing a careful review and constructive comments.

Reviewer #3 (Remarks to the Author):

Response: We commend Nature Communications' for its initiative to train Early-Career Researchers.

Reviewer #4 (Remarks to the Author):

Review of “Anatomy of a post-subduction collision” by Uluocak et al.

The manuscript by Uluocak and co-authors presents an interesting study on the relationship between mantle thermal structure and various observables from the Arabian–Eurasian collision zone. Overall, the paper is well written and addresses a topic of broad relevance. The potential implications are significant. However, I believe the current modeling approach requires substantial strengthening before the conclusions can be fully supported.

Major comments

1) Modeling strategy

-To my understanding, only a single simulation is presented. This limits the robustness of the conclusions. Sensitivity tests exploring variations in the initial thermal structure (inferred from geophysical constraints) are essential. Similarly, the role of mantle rheology, crustal architecture, inherited sutures, and other weak zones should either be explicitly investigated or at least discussed in detail.

-A direct comparison between model outputs and observations is missing. For example, modeled surface velocities should be plotted against GPS data in a single figure. Such comparisons are crucial to convincingly demonstrate the model’s validity.

-L328-L330. This is rather difficult to see both modelled velocities and GPS velocities are not super-imposed on a figure. Having a figure showing how well the model and the data fit would really help to convince the reader.

Response: We thank the reviewer for these important comments. We acknowledge that the manuscript presents a single numerical model to support its conclusions. This approach was necessary due to the space constraints of Nature Communications. However, the chosen model is the culmination of an extensive testing process, including comparisons with re-designed models from literature, the details of which we have already discussed in our previous studies (i.e., sensitivity of the model). Besides, as detailed in the Results and Discussion, our modeling results are strongly corroborated by independent datasets, including a new seismic tomography model and various geophysical and geodetic data, each sensitive to different physical properties (e.g., *Discussion*: *The calculated mantle flow vectors align closely with observations, such as modern kinematics (Fig. 1a), seismic anisotropy orientations and a low seismic velocity zone (Fig. 1c) that extends to the East African plume-magmatic belt at large-scales (Ershov & Nikishin, 2004; Civiero et al., 2023). These results reconcile previously disparate observations - from shallow (< 300 km), fast seismic velocities (at ~250 km depth; Koulakov et al., 2012) to overlying slow seismic anomalies (e.g., Koulakov et al., 2012; Civiero et al., 2023; the plumelet Uluocak et al., 2024)- into a unified geodynamic framework validated by different numerical models (Uluocak et al., 2021) based on recent seismic tomography data (Kounoudis et al., 2020).*”).

Our modeling strategy, as highlighted by Reviewer 2, further, involves correlating the results of our 3D thermomechanical model with key geophysical observables (i.e., direct comparison between model outputs and observations). We present a systematic analysis, for the first time in the region, showing positive correlations between high V_p/V_s ratios and magmatic arcs with highlands; between cold mantle anomalies and regions of thick lithospheric roots/deep basins, and between estimations and observed seismicity patterns. It is the strength of these consistent relationships that validates our model predictions and forms the core evidence for our interpretations. In response to Reviewer-4, we have improved the Method section to clarify validation of the numerical model:

Method: “Since different numerical models with various temperature configurations and rheological features were tested (Uluocak et al., 2021; 2024), in this study we followed a convection modeling approach with inputs that have already been confirmed/validated by considering large sets of geophysical data and analyses documented in the Arabian-Eurasian collisional region.”

We evaluated the Reviewer’s suggestion to include a direct comparison of surface velocities with GPS data, which would require further discussion on spatially resolved geodetic measurements in the region. However, given the stringent space and reference limitations, we have chosen to focus the manuscript on the core geodynamic analyses. We believe the existing correlations presented between our model results, major kinematics of plates, seismicity, and other geophysical observables provide robust and sufficient evidence for our arguments. Therefore, we feel that adding such a figure with references and discussion, while potentially interesting, is not essential to the central conclusions of this study.

2) Model setup and methodology

- The modeling approach is not sufficiently detailed in its current form. Model dimensions and resolution should be specified.
- Even in instantaneous flow models, runs should be evolved long enough to achieve isostatic equilibration. What timescales were used?
- Initial conditions must be clearly introduced: what is the starting thermal and density structure?
- A table summarizing all material properties should be provided, ideally in the appendix.
- What is your model size and resolution?
- You also need to provide a table with all used material parameters.

Response: We thank the reviewer for their comments on expanding the explanation of the model. Following the suggestions, we have added a new Table 1 to the revised manuscript that lists the material parameters. To accommodate this, we have removed some references from this section (Glerum et al., 2018; Uluocak et al., 2016, 2019) and added the relevant ones to the table (Ranalli, 1995; Naliboff & Buiter, 2015). Furthermore, we have improved the Methods section with additional explanations regarding the model's spatial resolution, time scale, and the initial condition of the model.

Method: “...We used seismic data (Piomallo & Morelli, 2003) to constrain the geometry of lithospheric structures and mantle features for the model (Uluocak et al., 2024: Uluocak, 2024a). Since estimations of instantaneous thermomechanical models are data-dependent, sensitivity analyses for the modeling approach used here were performed and documented in previous work on the collisional zone ...The temperature model (e.g., Figs. 3b, 4d & 5e) was obtained by using seismic data with a density scaling approach based on the thermal-expansion expression: $\rho(T) = \rho_0(1 - \alpha \Delta T)$ (reference densities, ρ_0 , are listed in Table 1; a coefficient of thermal expansivity, $\alpha = 2 \times 10^{-5} \text{ 1/K}$ was used) with the sampling intervals of 20 km in each direction for the investigated region (Uluocak et al., 2024). The time-stepping in the model was carried out using a Courant–Friedrichs–Lewy (CFL) number of 0.1, yielding a timestep of up to $\sim 1 \times 10^5$ years and relatively small velocity errors (i.e., $\text{RMS} \leq 2 \text{ cm/year}$) with the effective computation time (Uluocak et al., 2021).”

3) Figures and presentation

Figure 2: The contours are unclear. What variable is being contoured? This needs clarification.

Response: Based on Reviewers' suggestions, color scales, and missing location names have been renewed with an additional map to make more visible of deformation patterns in revised Figure 3 (previous Fig. 2). The caption has also been improved to more clearly indicate which variables are shown:

Figure 3. “Iso-surfaces are imaged only for ≤ 1000 K (see color scale in Fig. 4d).”

-Figure 3A: The method used to calculate dynamic topography must be described.

-Fig. 3a. How do you compute the dynamic topography? This is quite critical to explain how this is computed in your model as this is not trivial especially for non-modelers.

Response: We have improved the revised text with additional explanation of how we calculated dynamic topography to make it more accessible to non-modelers and the broad readership of Nature Communications. Due to journal space limitations, we also refer readers to specialized studies for a more detailed discussion of the modeling techniques. That is, we used the traditional approach to obtain the surface deflection of the top of the model solution space as reported in previous studies:

Method: “Dynamic support in the topography- dynamic topography (e.g., Faccenna et al., 2014)- is defined along the free surface based on the vertical stress exerted on the lithosphere as a response to normal fluid stresses in the underlying mantle.”

4) Data availability.

If I recall correctly, Nature Communications requires that the data necessary to reproduce the models and generate the figures be made publicly available, for example via Zenodo.

Response: Agreed. ASPECT is an open-source code available to the general public. As is standard in publications with models, our input files and ASPECT (with open-source functions in ASPECT) can be compiled from Uluocak et al., (2024) and Bangerth et al. (2022) as cited in the manuscript. Seismic data used in modeling is also available in Piromallo & Morelli (2003), as stated in the text (e.g., Introduction, Method). Besides, following the reviewer's suggestion, we have enhanced the Method and Data Availability sections with an additional reference (Uluocak 2024, below) to clarify accessibility and complicity of our modeling input code via Zenoda, ensuring the same level of openness as the ASPECT code itself.

Uluocak, E. Ş. (2024). Dataset for “The role of upper mantle forces in post-subduction tectonics: Plumelet and active rifting in the East Anatolian Plateau” (G-Cubed). [Dataset]. Zenodo. <https://doi.org/10.5281/zenodo.12568287>

In accordance with our commitment to open science, the corresponding author (ESU) will provide assistance to researchers seeking to compile the data upon reasonable request. The Data Availability Statement has been now updated to the following standard explanation from Nature Communications to formally confirm our compliance:

Data available on request from the authors

The data that support the findings of this study are available from the corresponding author upon reasonable request.

Nature Communications
Nature Neuroscience

<https://www.nature.com/documents/nr-data-availability-statements-data-citations.pdf>

Data availability: “Open-sourced functions/codes and data used in the numerical model can be compiled from Uluocak, (2024) that are publicly available. The data that support the findings of this study are available from the corresponding author upon reasonable request.”

-The manuscript presents a substantial body of work and demonstrates the authors’ strong understanding of the Arabian–Eurasian collision zone. However, the modeling component requires further work and sensitivity testing to ensure the robustness of the conclusions. Addressing these points would significantly strengthen the study and enhance its impact.

Response: We would like to clarify that a sensitivity analysis for a similar modeling approach was performed in previous work (Uluocak et al., 2021, 2024). The current manuscript expands on that foundation to provide a deeper insight into deformation mechanisms in collisional settings. The Method section has been revised to explain this:

Method: “We used seismic data (Piromallo & Morelli, 2003) to constrain the geometry of lithospheric structures and mantle features for the models. Since estimations of instantaneous thermomechanical models are data-dependent, sensitivity analyses for the modeling approach used here were performed and documented in previous work on the collisional zone (Uluocak et al., 2021, 2024).”

Line per line comments:

L22. I am not quite sure to know/understand what is a “plumelet” at that stage (I am not on top of this literature). As this is likely a key concept of the paper you may want to clarify this for a general audience, right from the start. Is that a small plume? Is it super-adiabatic? Is its buoyancy chemically controlled? (*in Abstract: collision—an archetype of post-subduction tectonics. Our key findings reveal that plumelet-plate interactions drive deformations both within and at the margins of convergent plates.*)

Response: To avoid repeating our earlier work where the plumelet concept was first established in this region, we provide a succinct explanation -not in the Abstract but -in the text:

Tectonic context and motivation: “...including a large-scale low-velocity zone extending from Afar to the Greater Caucasus ($\leq \sim 300$ km depth, Fig. 1c, e.g., Civiero et al., 2023), the latest numerical model by Uluocak et al. (2024) implies a lithospheric channel with SW-NE mantle flows (Ershov & Nikishin, 2004) — a plumelet (i.e., upper-mantle plume migration without significant tail and a mushroom head; Uluocak et al., 2024)—here.”

L54. “and the geodynamic model of the upper mantle and lithosphere structures”. Is that based on previous work and reconstructions? You may need a reference here. (*“the geodynamic model of the upper mantle and lithosphere structures, we resolve the 55 three-dimensional (3D) mantle circulation patterns, including large- and small-scale convective cells, and their dynamic interactions with overlying plates. Different from previous studies, 57 including numerical experiments in this part of the collision (Fig. 1b, e.g., Faccenna et al., 2014; 58 Uluocak et al., 2021; 2024 and references therein),*

Response: Thank you for the reviewer, this part has been rephrased to avoid misunderstanding as follows:

Introduction: “Based on seismic imaging (Piromallo & Morelli 2003) and our geodynamic model of the upper mantle structures, we resolve the three-dimensional (3D) mantle circulation patterns with large- and small-scale convective cells, and their dynamic interactions with overlying plates.”

§ L51-L65. It is not clear how the model is built from geophysical and geodynamic data at that stage.

(51 Here, we focus on the Arabian-Eurasian collision region (Fig. 1) as a plate collision 52 archetype where a multitude of post-subduction mantle dynamics are actively shaping the 53 surface geology and tectonic configuration. Based on seismic imaging (Piomallo & Morelli 54 2003) and the geodynamic model of the upper mantle and lithosphere structures, we resolve the 55 three-dimensional (3D) mantle circulation patterns, including large- and small-scale convective 56 cells, and their dynamic interactions with overlying plates. Different from previous studies, 57 including numerical experiments in this part of the collision (Fig. 1b, e.g., Faccenna et al., 2014; 58 Uluocak et al., 2021; 2024 and references therein), our 3D thermomechanical model, extending 59 east to the Caspian Sea, is sensitive to both large (e.g., plumelet, Uluocak et al., 2024) and 60 regional-scale changes, which may correspond to specific zones with thinned/missing 61 lithospheric mantle (e.g., the proposed delamination-modified regions of eastern Anatolia and 62 western Greater Caucasus), high plateaus (e.g., Turkish-Georgian-Armenian-TGA, the East 63 Anatolian and the NW Iranian plateaus), deep basins (e.g., Mesopotamian-Zagros, Kura, 64 eastern Black and western Caspian seas basins) and plate boundaries (e.g., Bitlis-Zagros suture 65 and fold-thrust belt) (Fig. 1).)

Response: According to the journal's format we introduce our aim, approach, and results briefly in the Introduction and expand more in the **Method**. And as cited from the various sources, we use geophysical data such as seismic to constrain the geometry of lithospheric structures and mantle features for the model. The citations from previous studies reference the basis for various configurations and assumptions we make for the model. In the revised manuscript we have made a number of changes in response to specific points from all three Reviewers to clarify aspects of the setup and design of our model. We believe that these citations and expanded explanation together provide a clearer explanation of how the model was built.

L68-L69. "By coupling mantle flow models with thermal 68 and isostatic constraints" this partially answer my previous comment. Maybe this should be moved in previous section.

(67 Our comprehensive geologic-geophysics-geodynamic analyses illuminate the tectonic 68 anatomy of this complex post-subduction system. By coupling mantle flow models with thermal 69 and isostatic constraints, we reveal a SW-NE oriented plumelet from the Arabian foreland 70 injecting into the subduction system, 3D thermomechanical erosion of the upper plates and the 71 active Neotethys slab deformation along the southern collisional front.)

Response: According to the journal's format we introduce our aim, approach, and results briefly in the Introduction, respectfully we choose to retain this flow.

L73-L74 . "Our results for the first time establish new constraints", remove "for the first time", as you have "new constraints" in the same sentence, and also because you use "for the first time" L78.

"73 Our results for the first time establish new constraints on drip-74 like removal beneath the post-orogenic magmatic arc under 75 convective forces. Compared to the well-studied seismotectonics beneath the Greater Caucasus 76 Range, the poorly defined tectonic implications of crustal and subcrustal earthquakes along the 77 Arabian collision front between the Arabian-Anatolian-Iranian plate boundary are clarified for 78 the first time in this study."

Response: Thank you for the suggestion, it has been done.

Fig. 1c. Shouldn't the caption be ΔV_s (or ΔS -wave and not just S-wave) as you display % variation? I am not a geophysicist, but S-wave should be in km/s. If you have % it likely means that this is a variation from a reference model.

Response: Both versions can be accurate, we utilize the same units as in the original study-Civiero et al. (2023).

L147-148. “The northern branch of the slab subducted beneath Eurasia by creating arc and back-arc volcanism”. Something is wrong in this sentence. I suppose you meant “by creating” -> “created”?

“147 The northern148 branch of the slab subducted beneath Eurasia by creating arc and back-arc volcanism, mostly149 dating from the Early Cretaceous to Eocene”

Response: The Reviewer is correct. We have fixed this grammatical error to “The northern branch of the slab subducted beneath Eurasia created back-arc volcanism, mostly dating from the Early Cretaceous to Eocene.”

L165. I don't see the Erzurum-Kars plateau “T” on figure 1D.

(165 the Erzurum-Kars Plateau (“T”; Fig. 1d))

Response: The letter 'T' in Figure 1d indicates the location of the Erzurum-Kars Plateau to simplify and make clear the image. The letter T in the revised Figure 1d was bolded, and the text was clarified as follows:

“the Erzurum-Kars Plateau (locations shown by “T” in Fig. 1d)”

L192-L195. This should go in method and be expanded (and partly moved to appendix too). How did you integrate the data? Did you invert solely for temperature and you that to define the thermal structure of the model?

(192 We determine the regional mantle circulation and lithosphere deformation with a 3D

193 instantaneous thermomechanical model based on seismically defined upper mantle structures194 (Fig. 2).

The results define the anatomy of the structures and dynamics that interact in a complex195 way in this active post-subduction system of the Arabian-Eurasian collision zone.)

Response: We employ a modeling method previously developed and discussed (Uluocak et al., 2021, 2024). We thank Reviewer 4 for their comment and confirm that the inversion was indeed performed only for the temperature field to define the model's thermal structure. It is also important to note that the core objective of this study is not to present a novel code/modeling approach, but to utilize proven and robust methodology to investigate complex deformation mechanisms. In response, we have revised the manuscript to make this modeling strategy more explicit for a broader audience in the Method section.

What are the densities of your material? How do you choose them? Are they computed from phase equilibrium modelling? How sensitive is you modeling results when propagating error of the inversion? Are you only imposing the initial temperature structure? What are the assumptions in terms of rheologies and material properties for the different crustal and lithospheric units?

L298. The temperature model should be presented way earlier in my opinion as this is the basis of your modeling!

Response: As suggested by Reviewer 4, Table 1 and additional references have been added to the revised manuscript, and this includes material properties, including density. Model sensitivity was already discussed by our previous papers cited clearly in the text (and ASPECT is benchmarked, as described in these references). As we do not introduce new code in this work, and also because of space limitations, there is not an opportunity to repeat such discussion here. We have expanded description of the temperature model in the revised Figure 3 (former Fig. 2); at a stage well before Figs. 5 and 6 (modelling results). The Method section has been revised as follows for further description:

-Variation in the temperature model should also be explored to test how sensitive the modelling outcome is.

Response: Because of space and reference constraints, we could not include all previously tested models for this collisional zone. Our aim is to focus specifically on deformation mechanisms, having already explored various parameters and thermal configurations in prior work. In response to the Reviewer's comment, we have expanded the revised text with additional explanation concerning the modeling methodology (see response above, Method and Results in the tracked change file). The full suite of input parameters is detailed in the following figures from our previous studies: Figure 5 in Uluocak et al. (2021), Figure 3 in Uluocak et al. (2024), and Figure 6 in Uluocak et al. (2021). We have cited relevant references clearly in the text for detailed input parameters.

-Do you define fault, shear zones and weak zones? How are they distributed? Although it does not look like you impose such structures, how important can they be in controlling the stress-strain state of the region? I would suspect that they are rather important. Moreover, I understand you perform instantaneous flow models. Do you account for elasticity? How is isostatic equilibrium achieved? How many timesteps are performed to reach elastic and isostatic equilibrium?

Response: We did not impose any pre-existing weaknesses in the model that could cause deformation, such as shear zones. Any inputs from a priori knowledge should be important in evolutionary models, but this is not the case here. Besides, we don't consider elasticity, mass transfer, and radioactive heating. The Method section has been improved to more clearly define our approach and modeling criteria, and to clarify possible misunderstandings raised by the reviewer, as follows:

Method: "The time stepping in model was carried out using a small Courant–Friedrichs–Lewy number of 0.1, yielding a timestep of up to $\sim 1 \times 10^5$ years and the relatively small-velocity errors (i.e., $\text{RMS} \leq 2$ cm/year) with the effective computation time (Uluocak et al., 2021).....Any possible responses of elasticity, mass transfer, and radioactive heating were not considered in the incompressible solution domain. Since different numerical models with various temperature configurations and rheological features were tested (Uluocak et al., 2021; 2024), in this study we followed a convection modeling approach with inputs that have already been confirmed/validated by considering large sets of geophysical data and analyses documented in the Arabian-Eurasian collisional region. For a more detailed discussion of the modeling methodology and parameters, the reader should refer to Uluocak et al., (2024) and Bangerth et al. (2022)."

L197-L199. At that stage I am still not quite sure to understand what is a plumelet. It seems to me that this is simply a buoyancy contrasts driving upward motion. An upwell?

"197 Mantle dynamics in the region are dominated by a focused rapid SW-NE directed inflow198 from the Arabian foreland to the eastern Greater Caucasus—a plumelet— penetrating through199 the complex morphology of the post-collision environment (Fig. 2)."

Response: A plumelet can be explained as a regional upper-mantle plume migration with neither a significant tail extending to the lower mantle nor a mushroom head reaching the hot spots on the surface as stated in the Tectonic context and motivation before Results:

Tectonic context and motivation: ... " a plumelet (i.e., upper-mantle plume migration without significant tail and a mushroom head; Uluocak et al., 2024)"

Fig. 2. What are the iso-contours you are showing on the figure?

Response: The caption of revised Figure 3 has been corrected to clarify the variables.

Figure 3.: “...Iso-surfaces are imaged only for ≤ 1000 K (see color scale in Fig. 4d).”

Figs 3b,c Shear stress and Strain rate. Are those invariants? Second invariant the shear stress and strain rates? Otherwise they should be tensors...

Response: Agreed, we thank for the correction; the revised text has been improved by corrected definition of parameters:

Revised Fig. 4: “(b) Amplitudes of the shear stress in a slice at 20 km depth”

Method: “...the square root of the second invariant of the strain rate tensor...
...and the shear stress tensor...”

Table 1.: “... the square root of the second invariant of the strain rate tensor is shown as strain rates (1/s) for simplicity in the study.”

Fig. 3d. 3D temperature model with a slice. Are you plotting an iso-surface of the temperature? + a slice? Moreover, there are no yellow dashed lines in Fig. 3d. It needs be to moved in caption 3c.

Response: The caption has been corrected, thank you for the comment. In Fig. 4d, we imaged the 3D temperature field for a specific depth range ([40-200] km with a temperature slice at a specific depth of 200 km.

Fig. 5. Is that realistic to have temperature reaching 2000°C at 300 km depth? Also it seems that large regions of very low temperature are predicted at great depth $\sim 1000^\circ\text{C}$. Without contouring I have trouble to exactly see the temperature structure, but it seems rather extreme.

Response: For unity, we use scientific coloring for the same features in the text. It is suggested that average temperature can reach 1628 K at 100 km depths in the Anatolian Plateau (e.g., Artemieva, & Shulgin, 2019, <https://doi.org/10.1029/2019TC005594>). These extreme temperatures can be linked to lithospheric deformation mechanisms and plumelet-induced thermomechanical erosion, as we claim in this study. We thank the Reviewer for this comment; the method has been improved with additional explanations:

L279-L284. Where is the cold anomaly on the figure 3d. I don't see it. Actually, the figure shows really well the trench parallel hot anomaly due to slab break off.

Response: We thank the reviewer for the interpretation although we are not sure we can understand how they reached this conclusion. The text has been corrected as follows:

Results: “arc-parallel relatively cold anomalies (< 1500 K) beneath the Transcaucasian zone...”

L288-L290. The “Sinking British Slab” of figure 5b?

(288 In the active Arabian collisional front, the results discover an interaction between the 289 plumelet and cold upper mantle structures, namely a mantle anomaly beneath the thick northern 290 Arabian lithosphere (~ 250 km depths; Fig. 5b).)

Response: We don't think we fully understood the question here; the very next sentence explains the Bitlis slab anomaly. This part has been improved by the previous Reviewers' suggestions (see above) and these are documented in the tracked change file.

L301-L305. First, this slab sinking and related upwell and potential erosion are very close to what seems the boundary of the model. How is this constraining the results? Or is it because of how the figure is clipped?

Response: We aim to make visible all contrasts that might indicate deformations in the study

area by choosing a consistent color scheme and view of figures, including revised Figure 6 (former Fig. 5). Interpretations in the text regarding the deformation mechanisms (sinking-tearing-breaking of the slab, erosion etc.) do not change based on the full-sized cross-section shown below. We should note that this interpretation: “First, this slab sinking and related upwell and potential erosion are very close to what seems the boundary of the model” made by the Reviewer can be accurate based on their models, but we are not suggesting such a type of deformation here based on our model.

L337-L340. This is also difficult to visualize this based on figure 6. Would be better to have a figure super-imposing seismicity and modeled second invariant of the strain rate for instance.
Response: Revised Figure 7 (previous Fig. 6) has been improved in the revised text to clarify locations and seismicity by following the Reviewers’ constructive comments.

Fig. 6c Given that you are using non-linear T dependent rheology, the discretized viscosity colorbar is an odd choice

Response: We show this image in revised Figure 7c to emphasize the deformation patterns such as sinking Bitlis Slab, active plate tearing and mantle intrusion along the cross-section.

L349-L355. How can the southeastern propagation of the deformation be discussed in the light

of an instantaneous flow model? I may be misunderstanding here, but from my point of view this can only be explored when performing forward models.

Response: We appreciate the reviewer's perspective (i.e., interpretations provided in our work can only be done by forward models). We respectfully note, however, that the prevailing evidence from instantaneous modeling studies does not support this conclusion.

L390-L392. It seems like a bit of a circular reasoning here. As the small scale anomaly are the outcome of using the tomography model as input for the model.

(We identify small-scale cold anomalies that are also visible in independent seismic 391 tomography models beneath the eastern Pontides-Lesser Caucasus arcs and the East Anatolian 392 Plateau)

Response: This is not circular reasoning, but the result of the flow modelling (which is, in turn, based on the seismically-derived mantle structures) based on the data imaging upper mantle structures.

L435- Where is the plumelet coming from? What drives its ascent?

Response: This is an important question that has been/is being considered by many researchers; please see Fig. 1c and relevant explanation in the text, e.g., “In the tectonic anatomy of the complex plate boundary, during its migration from deep mantle source(s) in the south...” Further discussion is beyond our scope in this study and cannot be conducted without additional data, investigations, and analyses.

RESPONSE TO REVIEWERS' COMMENTS

Reviewer #1 (Remarks to the Author):

Dear authors and editors,

I have read carefully the new version of the manuscript and the corresponding answers and it seems the authors have carefully addressed all comments and suggestions given by the reviewers, so in my opinion this version is acceptable for publication in Nat. Comm.

with kind regards

Response: Thank you for your constructive and insightful comments that improve our manuscript. Best regards.

Reviewer #2 (Remarks to the Author):

We are pleased to see the extensive responses of the authors to the comments / recommendations that we provided for this manuscript (and to the other reviews). We are satisfied that the authors have addressed the points we raised in a thoughtful manner and, given the constraints of space and the need to respond to other reviews, we are satisfied with their improvements. We endorse the paper for publication if the editors see fit.

Response: Thank you for your contributions and comments on this manuscript. Sincerely yours.

Reviewer #3 (Remarks to the Author):

Response: We wish you success in your career. Best regards.

Reviewer #4 (Remarks to the Author):

Review of “Anatomy of a post-subduction collision” by Uluocak et al.

In the revised version of the manuscript, the authors have addressed all of my comments. The study represents a very interesting contribution, and I thank the authors for their careful and thorough revision. I have only one minor comment remaining. Regarding Figures 6b and 6d, I would like to clarify my comment from the previous review, which may not have been sufficiently clear. When I wrote, “First, this slab sinking and related upwelling and potential erosion are very close to what seems to be the boundary of the model,” I was questioning the extent to which the lateral boundary conditions may influence the model behavior near the margins. For instance, in Figure 6e, the flow in the top left and top right corners appears to rotate rapidly and become parallel to the left and right model boundaries. My question is whether this behavior reflects a boundary effect. If so, how does this influence or constrain the interpretation of the modeled flow patterns and associated processes near the boundaries? Nicolas Riel

Response: Thank you for your comment regarding the patterns in the corners of the cross-sections in Fig. 6. We agree that modeling results are dependent on chosen inputs including boundary conditions. Hence, to ensure our interpretations are robust and not unduly affected by boundaries or specific parameters used in the model, we have systematically validated/confirmed our estimates against multiple independent lines of evidence at different scales. These include various seismic tomography datasets, patterns of magmatism, geochemical data, and results from large-scale numerical models that encompass our study region. We believe this multi-disciplinary approach strengthens the reliability of our findings. Thank you again for your insightful comment.